# Vertical redistribution of principle water masses on the Northeast Greenland Shelf

**Caroline V. B. Gjelstrup** [1]✉, **Mikael K. Sejr**[2,3], **Laura de Steur** [4], **Jørgen Schou Christiansen** [5,6], **Mats A. Granskog** [4], **Boris P. Koch** [7,8], **Eva Friis Møller**[9], **Mie H. S. Winding**[10] & **Colin A. Stedmon** [1]

The Northeast Greenland shelf (NEGS) is a recipient of Polar Water (PW) from the Arctic Ocean, Greenland Ice Sheet melt, and Atlantic Water (AW). Here, we compile hydrographical measurements to quantify long-term changes in fjords and coastal waters. We find a profound change in the vertical distribution of water masses, with AW shoaling >60 m and PW thinning >50 m since early 2000's. The properties of these waters have also changed. AW is now 1 °C warmer and the salinity of surface waters and PW are 1.8 and 0.68 lower, respectively. The AW changes have substantially weakened stratification south of ~74°N, indicating increased accessibility of heat and potentially nutrients associated with AW. The Atlantification earlier reported for the eastern Fram Strait and Barents Sea region has also propagated to the NEGS. The increased presence of AW, is an important driver for regional change leading to a likely shift in ecosystem structure and function.

Advection of low salinity water and sea ice from the Arctic Ocean is a key characteristic of the Northeast Greenland Shelf (NEGS). Dominance of this low saline layer over much of the area enforces strong upper water column stratification[1] which, combined with its low nutrient content, is currently the main limiting factor for primary production in the region[2,3]. The supply of freshwater to the NEGS is changing due to acceleration of mass loss from the Greenland Ice Sheet (GrIS)[4], increased central Arctic Ocean freshwater content[5], increased Arctic sea ice melt, and changes in freshwater[6] and sea ice export, in the western Fram Strait[7,8]. Similarly, the air-sea exchange of heat just off the NEGS has greatly increased due to the westward retreat of the sea ice edge[9].

On the NEGS, cold and fresh waters of Arctic origin meet warmer saline waters of Atlantic origin, forming a pronounced front as they flow southwards along the continental slope with the East Greenland Current (EGC)[10]. The upper 150–200 m of the EGC consist of Polar Water (PW) formed within the central Arctic Ocean[10,11]. Warm waters of

Atlantic origin (AW) are found below[12], and are originally carried northwards with the Norwegian Atlantic Current (NAC) and West Spitsbergen Current (WSC; Fig. 1). About half of this flow continues into the central Arctic Ocean where it is transformed into Arctic Atlantic Water (AAW) via cooling and freshening[12]. The other half branches off westward within the Fram Strait as Recirculating Atlantic Water (RAW)[13], thereafter merging with the EGC to return southwards[12] (Fig. 1).

As opposed to the dynamic environment at the shelf-break and wider shelf area where advection and eddy activity can cause rapid modification of hydrographic conditions, shelf signals can be preserved within subsurface waters of fjords with long residence times[14]. The properties of these waters can provide a record for observing ongoing Arctic climate change. Greenland fjords serve two important roles in this context: Firstly, they are the main gateways through which GrIS melt enters the ocean; and secondly, their bathymetry and circulation regulate the propagation of warm oceanic waters to marine-

[1]National Institute of Aquatic Resources, Technical University of Denmark, 2800 Lyngby, Denmark. [2]Arctic Research Centre, Aarhus University, 8000 Aarhus C, Denmark. [3]Department of Ecoscience, Aarhus University, 8000 Aarhus, Denmark. [4]Norwegian Polar Institute, Fram Centre, 9296 Tromsø, Norway. [5]Department of Arctic and Marine Biology, UiT The Arctic University of Norway, 9037 Tromsø, Norway. [6]Environmental and Marine Biology, Åbo Akademi University, Fl-20500 Turku, Finland. [7]Division of Biosciences, Alfred Wegener Institute for Polar and Marine Research, 27570 Bremerhaven, Germany. [8]University of Applied Sciences, An der Karlstadt 8, 27568 Bremerhaven, Germany. [9]Department of Ecoscience, Aarhus University, 4000 Roskilde, Denmark. [10]Greenland Climate Research Centre, Greenland Institute of Natural Resources, 3900 Nuuk, Greenland. ✉e-mail: cvbgj@aqua.dtu.dk

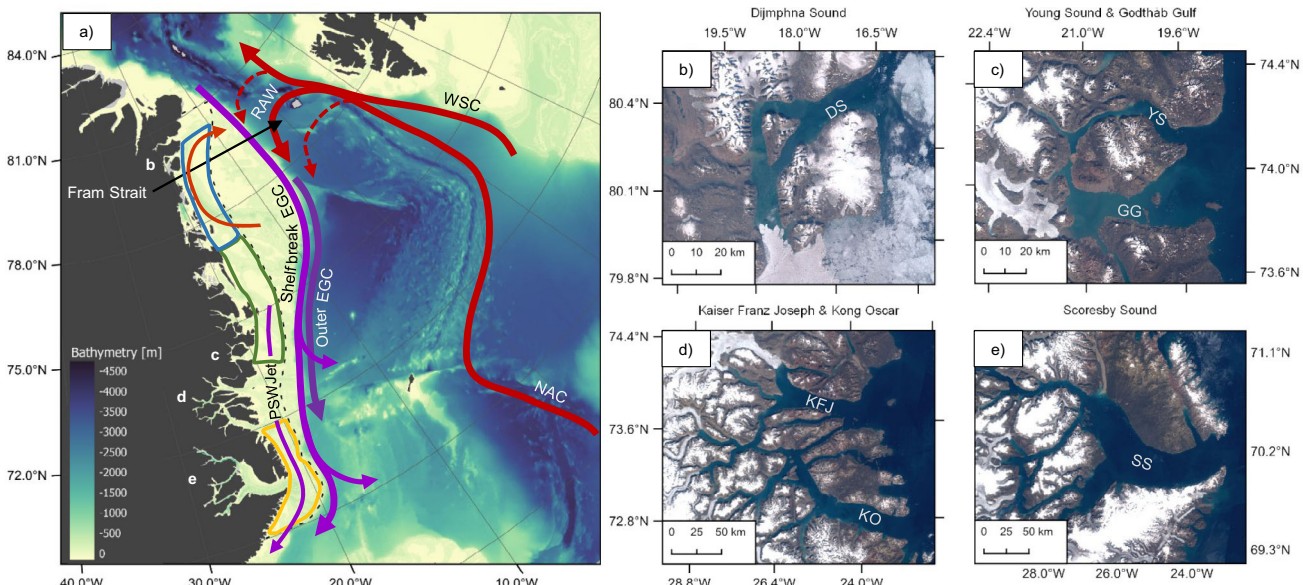

**Fig. 1 | Schematic of surface circulation in Northeast Greenland (NEG).** The main circulation features and water masses that influence the NEG shelf and fjords (**a**) are the East Greenland Current system (EGC) comprising of the Shelf break EGC, the Outer EGC and the Polar Surface Water Jet (PSW Jet) shown in purple, and the Norwegian Atlantic Current (NAC), the West Spitsbergen Current (WSC), and Recirculating Atlantic Water (RAW), shown in red. Currents are based on Håvik et al.[11] and augmented with near-surface shelf circulation (red current curving north and east on the shelf) from Bourke et al.[68]. Background shows bathymetry from the GEBCO 2020 product[65] where light shading represents shallow areas. The black dashed line indicates the near-shore region used for analysis of shelf waters throughout the current study. Blue, green and yellow polygons indicate the regions used in Fig. 5. Fjords included in the current study are shown in greater detail on panels **b**, **c**, **d** and **e**, on which the fjords have been labeled with their acronyms: DS - Dijmphna Sound, YS – Young Sound, GG – Godthåb Gulf, KFJ – Kaiser Franz Joseph fjord, KO – Kong Oscar fjord and SS – Scoresby Sound. From Copernicus Sentinel-2 image mosaic 2019.

terminating glaciers, contributing to mass loss through submarine melting and undercutting[15]. Whilst the mechanisms by which AW reaches the head of fjords are not yet fully understood[16], variability in coastal shelf waters is thought to be central in setting hydrographic conditions in the fjords[17]. In summer, the Northeast Greenland (NEG) fjords with deep sills that allows entrance of AW therefore have a three-layer structure similar to that found on the shelf: cold (temperature, $\theta < 0\,°C$), fresh (practical salinity, $S_p < 34.4$) PW advected from the Arctic Ocean comprises the intermediate layer and is capped by a thin, fresh local surface layer from summer melt and terrestrial runoff[18]. Finally, warm ($\theta > 0\,°C$), saline ($S_p > 34.4$) AW fills the deepest parts of the fjord. However, for fjords with shallow sills at the entrance, AW is inhibited from entering. The deep waters in these systems are therefore typically local modifications of the intermediate PW[18,19]. Note, that the detailed bathymetry of many NEG fjords and nearby shelf is still poorly known due to a dearth of observations.

Due to its sea ice coverage, the NEGS is relatively under sampled. There are two sustained long-term (>10 years) observation programs in the Young Sound fjord system (~74°N) and the western Fram Strait (~79°N). Results from the Young Sound monitoring program have shown a freshening in subsurface fjord and adjacent coastal waters over the 2003–2015 period[14,20]. Change has also occurred at depth where increased supply of heat from the passage of warmer AW to the 79N glacier has been documented[21,22] resulting in increased ocean induced melting of marine-terminating glaciers[23,24].

The distribution of PW and AW has important consequences for the shelf ecosystem across the Greenland shelf. Regions more influenced by AW generally show higher productivity due to the higher concentration of nitrate, while especially the NEGS has low overall productivity due to limited nitrate in PW and delayed spring blooms due to sea ice coverage[25]. Increased advection of AW or increased vertical mixing could directly increase the nitrate available to sustain phytoplankton production with potential cascading effects through the food web. Because PW and AW have different origin and

temperature, the two water masses also contain species with different biogeographic affinity. Increased abundance of boreal species on the East and West Greenland shelf have been linked to changes in water mass distribution[26–28]. The importance of changes in water masses for ecosystems is very apparent in the Barents Sea, where a distinct increase in AW presence has reduced sea ice cover, changed species distributions, and transformed ecosystem structure[29]. Since AW is also important for the NEGS we aim to answer if the Atlantification observed elsewhere has propagated to the NEGS. For this purpose, we consolidate previously disparate summer measurements of oceanographic conditions over the period 1923 to 2019 from the NEGS to reveal recent trends in selected fjords and place these in the context of hydrographic changes of the adjacent NEGS waters. Our analysis, which is restricted to waters up to 100 km from the coast, finds significant changes in core PW and AW properties and their vertical distribution during the last two decades. There are, however, distinct latitudinal differences of which the impact on upper ocean stratification is quantified and potential implications of these changes for the NEGS ecosystem are discussed.

## Results
### Changes in fjord water properties

Six fjord systems along the NEGS coast were studied (Fig. 1). Using consistent hydrographic metrics to characterize subsurface fjord water masses, individual fjords can be compared despite the range in size, bathymetry and hydrography that characterizes each system. PW ($S_p < 34.4$ and $\theta < 0\,°C$) and AW ($S_p > 34.4$ and $\theta > 0\,°C$) characteristics are reflected in the hydrography of the studied fjords (Fig. 2). The vertical temperature and salinity distributions in Dijmphna Sound, Kong Oscar fjord, Kaiser Franz Joseph fjord, and Scoresby Sound show stratified profiles showcasing the three-layered structure typical for Arctic fjords in summer. A fresh surface layer overlies the PW layer with characteristic sub-zero temperatures and relatively low salinity (<34.4). Salinity and temperature increase steadily below the PW layer

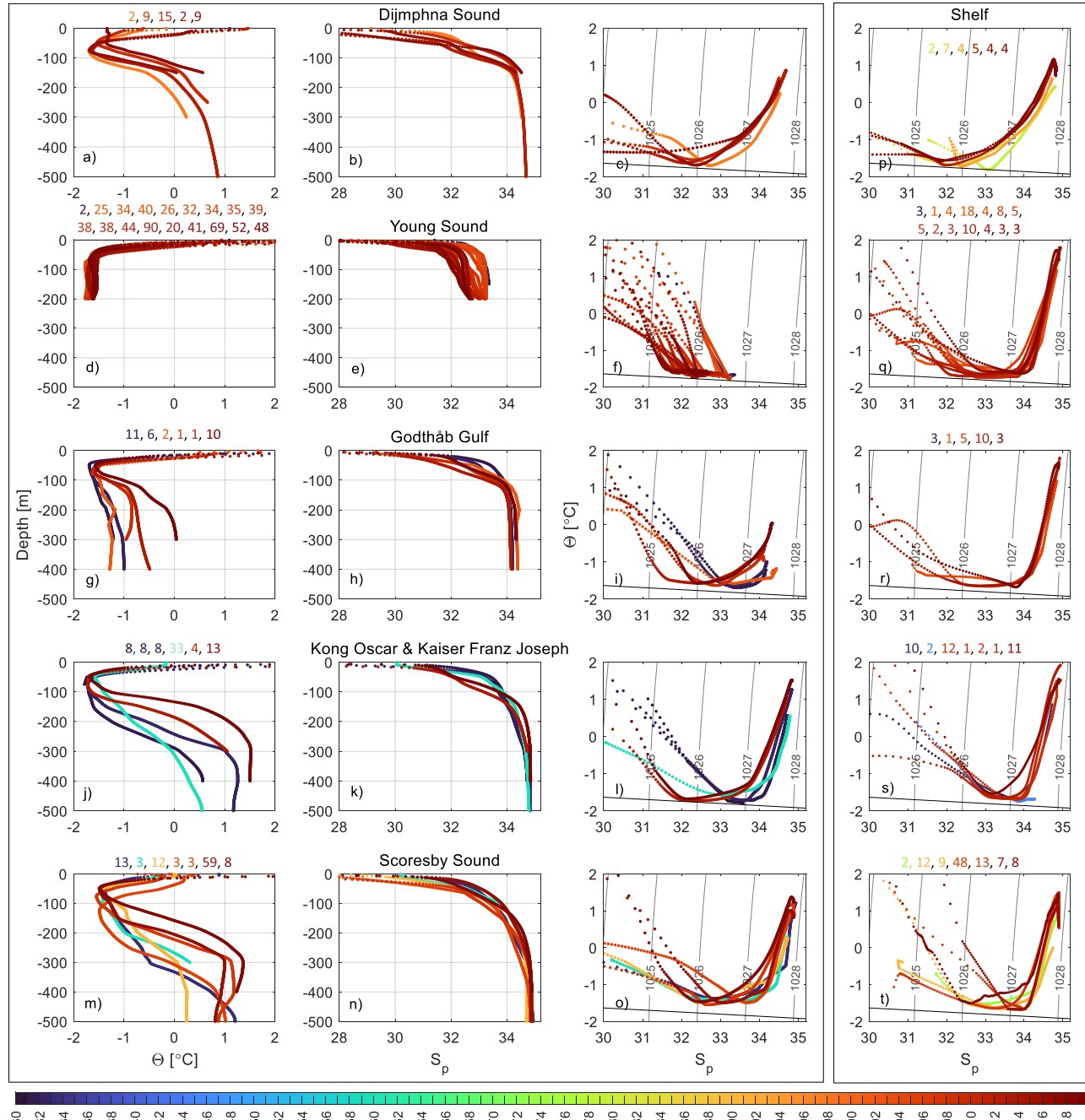

**Fig. 2 | Mean July/August/September profiles of temperature, salinity, and temperature-salinity (T/S) diagrams (first, second, and third columns) for the individual fjord systems included in the current study.** Dijmphna Sound (**a**–**c**), Young Sound (**d**–**f**), Godthåb Gulf (**g**–**i**), Kong Oscar and Kaiser Franz Joseph fjord (**j**–**l**) and Scoresby Sound (**m**–**o**). The raw data are shown in T/S space on Supplementary Fig. 3. Panels **p**–**t** (fourth column) show mean T/S diagrams of the shelf waters immediately adjacent to each fjord. Numbers above the panel give the number of profiles used for averaging and the sampling year is given by the color.

into the warmer and saline AW layer beneath. Shallow sills (45 m and 110 m deep) at the entrance to Young Sound and Godthåb Gulf hamper free inflow of AW, rendering a markedly different vertical structure. In these shallow systems, a thin fresh surface layer caps the strong gradients that occupy the upper 10–50 m. Below, PW dominates, and temperature and salinity change little with depth (Fig. 2d, e, g, h). In 2019 the bottom water temperature in Godthåb Gulf became positive, indicating the increased influence of AW at this time (Fig. 2g).

The properties of the PW and AW cores, defined as the temperature minimum and maximum within the two layers respectively, vary through the observation period (see Supplementary note 1 and

Supplementary Fig. 2 for how spatial variability within fjords was addressed). Whilst the magnitude and timing of change differs between the studied fjords, the general direction of change is similar. With the exception of Scoresby Sound at the southern extent of the region, all fjords undergo warming and substantial freshening of the PW core, clearly seen as a shift in T-S space (Fig. 2c, f, i, l). The most substantial PW core freshening is observed in Kong Oscar and Kaiser Franz Joseph fjords, where PW core salinity has decreased 1.53 from 33.84 in 1931 to 32.31 in 2018 (Fig. 2k), although the majority (97 %) of the salinity decline occurred between 1930 and 2013, decreasing further by only 0.04 between 2013 and 2019. PW layer in Young Sound has also freshened

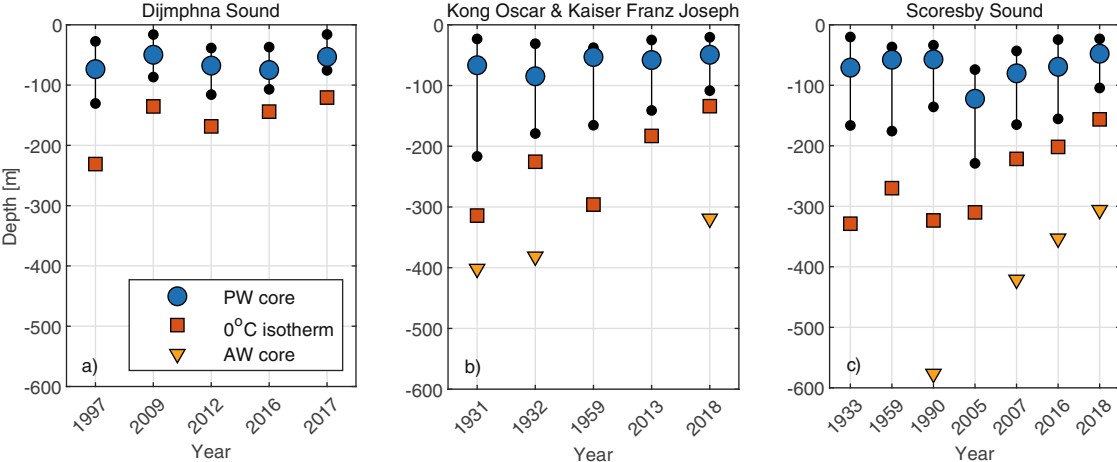

**Fig. 3 | Vertical distribution of water masses in three deep sill fjord systems during July/August/September.** Dijmphna Sound (**a**); Kong Oscar and Kaiser Franz Joseph fjords (**b**); and Scoresby Sound (**c**). The mean vertical placement of the Polar Water (PW) core (blue circles), the depth range occupied by PW (vertical black lines) and the depth of the upper Atlantic Water (AW) boundary (0 °C isotherm, red squares) for the three deep sill fjord systems. The mean depth of the AW core (yellow triangles) is shown for the profiles in which a distinctive temperature maximum was present. Note, the non-linear x-axis.

(linear trend of −0.10 y⁻¹, $p < 0.05$) over the 2003–2014 period. However, recently this trend reversed, and salinities have increased slightly (Fig. 2e).

Fjord summer surface waters across the region have also freshened. In Dijmphna Sound the mean salinity of the 0–20 m depth interval decreased by 3.11 between 1997 and 2012. This may, in part, be attributed to interannual variability and changes in sampling season, however, a similar decline in surface salinity occurred in Young Sound and Godthåb Gulf between 2003 and 2013, and Kong Oscar and Kaiser Franz Joseph fjords between 2005 and 2013. Scoresby Sound is an exception to this general freshening pattern, as no trends in surface nor PW core salinity are found.

Warming and salinification of the AW core is observed in all the deep fjord systems shown here. In Kong Oscar and Kaiser Franz Joseph fjords, the salinity of the AW core increased by 0.08 through the observation period (Fig. 2k), entailing considerable salinification from 150 m to the bottom. The AW core temperature was variable with a 0.78 °C increase observed between 1931 and 1932, which is similar in magnitude to the total warming observed (0.92 °C). AW core temperature also varied considerably in Scoresby Sound, where it increased from 1990 at a rate of 0.03 °C y⁻¹ to a mean value of 1.47 °C in 2018.

Even more pronounced is the change in the vertical distribution of PW and AW layers in the deep fjords (Fig. 3). The extent of the PW layer, defined as the thickness of the depth interval with temperature <−1 °C, in Kong Oscar and Kaiser Franz Joseph fjord has reduced dramatically from occupying almost 200 m in 1931 and only 90 m in 2018 (Fig. 3b). In concert, with a correspondingly substantial shoaling of the upper AW boundary, defined as the depth of the 0 °C isotherm, this represents a huge change in fjord hydrography and entails a large increase in fjord heat content by volume alone. Dijmphna Sound and Scoresby Sound likewise experience systematic shoaling of the upper AW boundary, amounting to an upward displacement of 76 m between 1997 and 2017 and 134 m between 1990 and 2018, respectively (Fig. 3a, c). Furthermore, shoaling of AW outside Godthåb Gulf was sufficient to clear the sill depth (110 m) by 2019, allowing inflow of AW and causing the increase in bottom water temperature (Fig. 2g).

## Regional scale shelf changes
The changes in fjord water properties correspond very well with observations from the adjacent shelf waters (Fig. 2, right panels). The overlap between fjord and shelf water properties (apparent as an overlap in T-S space) highlights the connection across the sills and shows that fjord waters are very much influenced by variability on the shelf. To place the observed fjord changes into a broader context, we analyze historical hydrographic data from the shelf going back to 1923. For this purpose, a time series of core PW and AW summer properties are constructed (Fig. 4; see "Methods" for how heterogeneous data distribution was considered). The analysis is spatially confined to the near-shore shelf region delineated by a polygon extending 100 km from the shoreline, unless the shelf is narrower, in which case the 500 m isobath is used (see Fig. 1, dashed black line). This delineation is made specifically to reduce spatial discontinuity in water mass properties across the shelf, as a more saline PW as well as a warmer AW are occasionally present when oceanographic fronts are traversed (Supplementary Fig. 4).

Although surface water (top 20 m) temperatures are variable (Fig. 4a), there is a systematic freshening evident from the mid-1990's through to the end of the time series (Fig. 4b). Over this period, the mean surface salinity dropped by 1.8 from 31.55 in 1996 to 29.74 in 2019. A similar pattern is apparent in the PW properties where there are no systematic changes in temperature (Fig. 4c), but an apparent freshening, however, the development in salinity differed with latitude (Fig.4d and Supplementary Fig. S5). North of 73.5°N the PW core freshened at a rate of 0.5 per decade between 2000 and 2019, with a mean salinity of 32.67 over the 2010–2019 period compared to a mean salinity of 33.35 over the 1995–2005 period (Fig. 4d). The salinity of the PW core in the southern region is more variable (Fig. 4d).

The properties of AW in the region have also changed with core temperatures increasing ubiquitously over the near-shore NEGS for the past four decades (Fig. 4e and Fig. S5). Currently, AW is >1 °C warmer than in the late 1990's. The record of AW salinity north of 73.5°N reveals salinification of 0.05 per decade between 1997 and 2019, while no systematic change was observed south of 73.5°N (Fig. 4f).

In addition to pronounced PW freshening and AW warming the vertical distribution of these waters have changed. The record of PW thickness shows that the PW layer is typically between 100 m and 200 m thick on the shelf but has been thinning from the 2000's and onwards (Fig. 4g). This thinning trend occurs at a rate of 22.6 m per decade between 2000 and 2019 and is accompanied by shoaling and thickening of AW. Whilst there is a clear AW shoaling trend, interannual variability through the time series is large (Fig. 4h). Since 2000, the 0 °C isotherm has shoaled by more than 60 m.

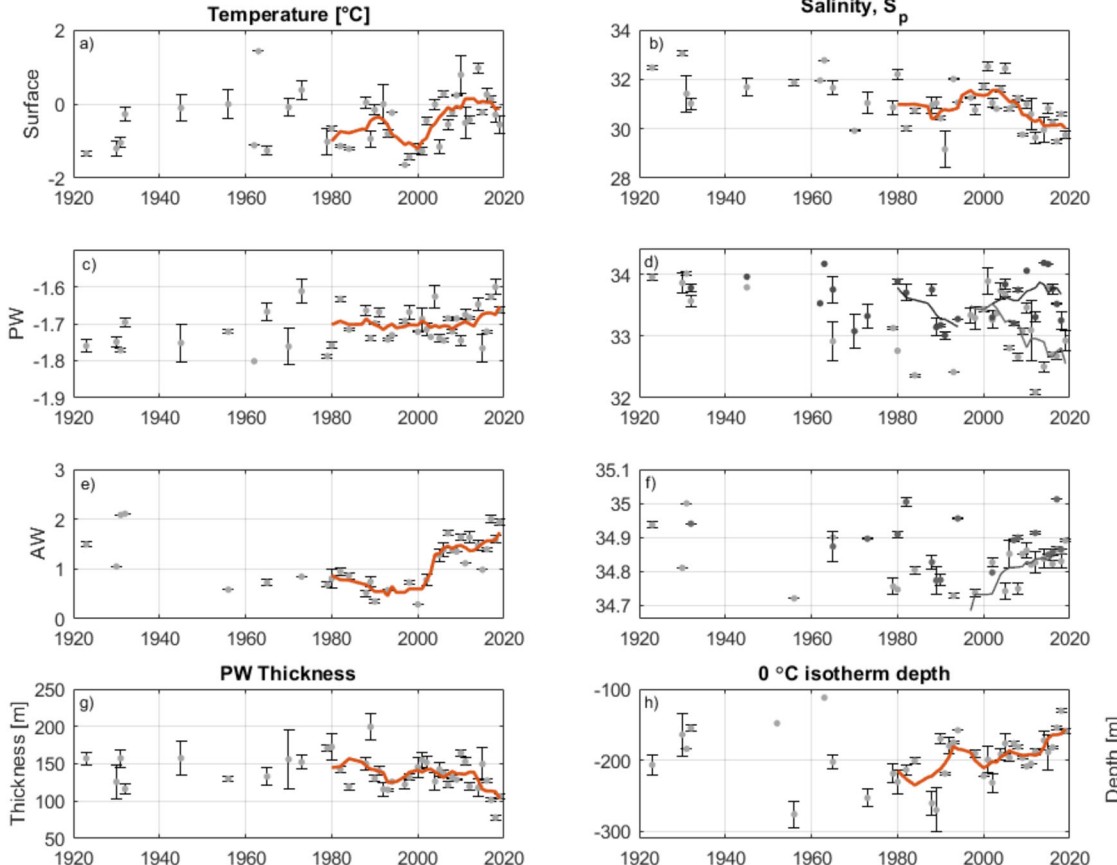

**Fig. 4 | Time series of summer (July/August/September) shelf water properties.** Surface water temperature (**a**) and salinity (**b**) defined as the mean of the 0–20 m depth interval, Polar Water (PW) core temperature (**c**), salinity (**d**), and layer thickness (**g**). Atlantic Water (AW) core temperature (**e**), salinity (**f**), and depth of the upper boundary of AW (**h**). In (**d**) PW and (**f**) AW core salinity is presented north (light gray) and south (black) of 73.5°N to show difference in trends. Division north and south of 73.5°N can be seen on supplementary Fig. 5 for all metrics. A 5-year running mean filter was applied to emphasize trends since 1980 (orange lines). Error bars indicate the standard error.

## Impact of hydrographic changes on stratification

The combined impact of the hydrographic changes on stratification can be evaluated through the concept of available potential energy (APE), which represents the energy required to break down vertical density gradients and bring a given water column into mixed state[30]. AW is particularly important in this context as it acts as a marine heat source for ice melt as well as an important source of nutrients for phytoplankton in the photic zone. APE is quantified from the surface to the depth of the upper AW boundary and is used to document changes in the accessibility of heat and nutrients in AW to surface waters over the NEGS. Changes in APE are dependent on density changes as well as interface depth changes (see Methods)[31]. As such, APE is a measure of the integrated effects of changes in stratification in the upper water column.

Three characteristic areas can be distinguished according to stratification; a northern region between 80°N and 77°N in which stratification is strongest (largest APE), a central region from 77°N to 74°N, and a southern region between 72°N and 69°N where stratification is comparatively weak (Fig. 5a). We note particularly sparse data coverage between 74°N and 72°N and therefore exclude this stretch of the shelf. A shift in time in this latitudinal trend of APE becomes apparent when splitting the time series into two periods, before (darker colors) and after 2004 (lighter colors), reflecting the onset of significant hydrographic change discussed above. APE is highest in the northern

region for the period before 2004 (mean 1318 J m⁻³) and has decreased 14% to 1130 J m⁻³ in recent years (Fig. 5a). In the central region, APE is lower but very variable and as a result shows no significant change over time. In the southern region, APE has decreased 35% from 1179 to 765 J m⁻³ after 2004. The APE changes can be attributed to: shoaling of the 0 °C isotherm; changes in density; and a cross term which represents the non-linear interaction term. The results show that the large reduction in APE in the southern region is primarily due to upward displacement of AW (Fig. 5b), and to a lesser degree due to density changes (Fig. 5c). Similarly, shoaling of AW drives much of the decline in APE in the northern region, however, simultaneous freshening of the surface and PW layers counteracts this effect (Fig. 5d). The latter effect is however relatively poorly constrained as reflected by the change in sign of the density contribution when one standard deviation around the mean is included in the calculation (Fig. 5d, error bars).

In contrast to the northern and southern reaches of the shelf, the central region experiences no significant change in APE between the two periods (Fig. 5a) due to the large variability in the data from before 2004. There is a combination of thickening (Fig. 5b) and freshening (Fig. 5c) of the PW layer. Thus, the nutrients and heat associated with the AW remain well isolated from the surface here. The overall analysis reveals that density changes in the surface and PW layers, and AW shoaling, have altered the latitudinal pattern of stratification along the near-shore NEGS. A schematic summarizing these physical changes is

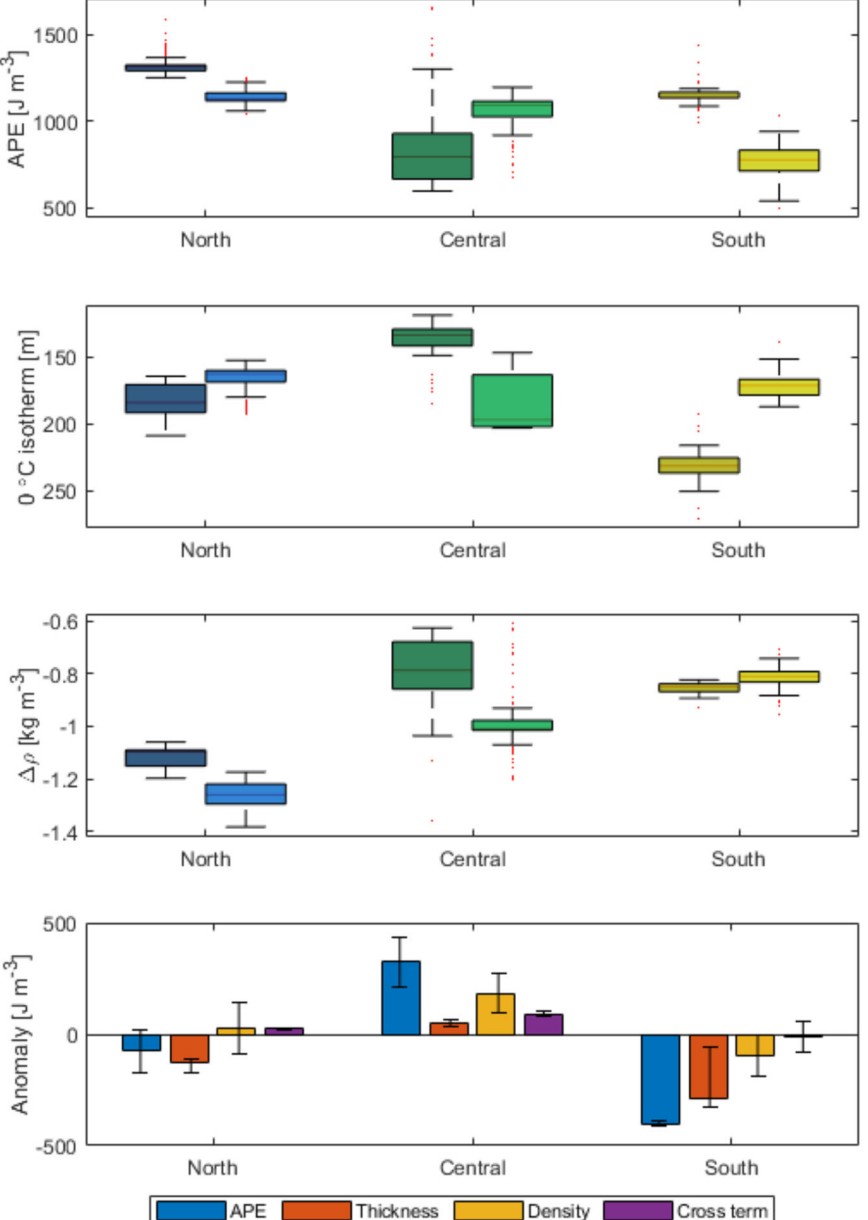

**Fig. 5 | Changes in Available Potential Energy and its constituents during July/ August/September.** Available potential energy (APE, **a**), depth of the 0 °C iso-therm (**b**) and mean density deviation (Δρ) from the surface to the depth of the 0 °C isotherm (**c**) in the north (81°N–77°N), central (77°N–74°N), and south (72°N–69°N) regions over the near-shore shelf. The dark and light colour shading give the 1980–2004 the 2004–2019 periods, respectively. Box plots indicate the median (middle line), 25th and 75th percentile (box) and the most extreme data points that are not outliers (box whiskers). The bottom panel shows the APE anomaly between the two periods and the thickness, density and cross-term contributions hereto (**d**). The error bars indicates one standard deviation.

provided in Fig. 6 and will be discussed in connection with knock-on effects for the ecosystem in the following discussion.

## Discussion

There is evidence for a significant summertime freshening of the sur-face and PW layer over much of NEGS. Potential sources are GrIS melt water and freshwater exported from the central Arctic Ocean via Fram Strait in either liquid or solid state. Although the Northeast GrIS appears to have been in equilibrium up until the early 2000's[32], acceleration and thinning at the ice sheet margins has since then, accelerated mass loss to the coastal environment[33]. The total freshwater flux from Northeast Greenland has nearly doubled from $125 \pm 9$ km³ y⁻¹ between 1992 and 2010, to 240 km³ y⁻¹ from 2007 to 2016[34,35]. Stable isotope and dissolved organic matter measurements have shown that freshening of NEGS waters due to GrIS melt is largely

limited to surface waters with salinities less than 31.5[36]. So it is possible that GrIS melt has contributed to the decline in surface water salinities, but not the decline in PW salinities.

Liquid freshwater export from the central Arctic Ocean via Fram Strait is composed of contributions from sea ice melt, river discharge, precipitation and less saline ocean water entering through the Bering Strait[37]. Whilst the composition in terms of source waters of the Fram Strait outflow is highly variable, evidence for increased positive sea ice melt water contribution to the total freshwater content since 2009 has been reported[38]. Freshwater transport within the shelf-break EGC showed no trend between 1997 and 2009 (-1300 km³ yr⁻¹[39]) but increased between 2010 and 2015, amounting to an additional fresh-water volume of 3106 km³ [4,40]. While this may have caused some freshening on the NEGS, the bulk has likely followed the shelf-break EGC southward to Denmark Strait as the fresh anomaly was also

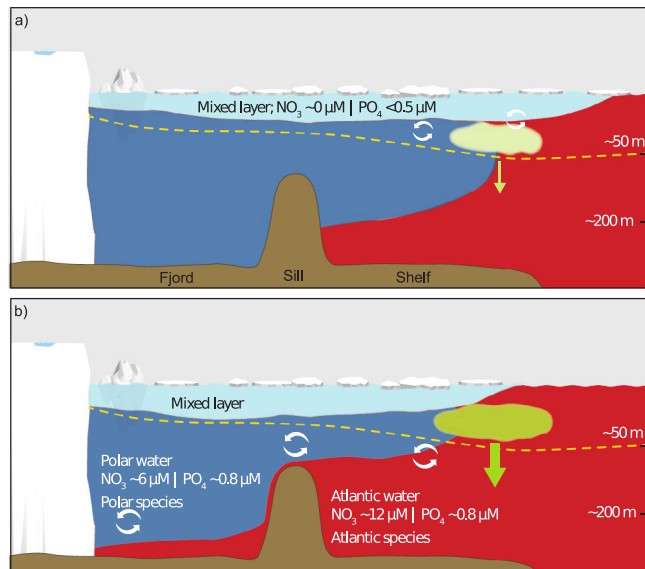

**Fig. 6 | Schematic of the changes occurring in summer vertical stratification and potential impact of phytoplankton production on the shelf.** The transect is a schematic of conditions within fjords (left) and eastward across the shelf (right of the sill). **a** Summer conditions where surface waters (light blue) become stratified and are nutrient depleted due to phytoplankton growth and nitrate is the limiting nutrient[1,49]. Nutrient supply to surface waters is from entrainment from water below which is either Atlantic Water (AW) or Polar Water (PW) and this supports a subsurface phytoplankton community. The seasonal retreat of sea ice exposes surface waters in the frontal region to greater potential of vertical mixing. **b** Evolving summer conditions with thinning of the PW layer and further sea-ice retreat, resulting in lower energy required to mix nutrients from AW upwards and deeper penetration of light, and as a consequence potential to sustain higher phytoplankton productivity and increased export to benthic biota. The yellow dashed line indicates the photic depth, which with a thinning of the PW layer can be expected to deepen and potentially extend further into AW[50].

observed here with little delay. However, since the freshwater transport with the PSW Jet on the 300 km wide northern NEGS is not monitored year-round, a liquid FW contribution from the Arctic Ocean through a pathway on the shelf to the observed freshening on the shelf cannot be ruled out. Recent findings from further east on the NEGS report a reduction in Polar freshwater export through the Fram Strait since 2015, show a thinning of PW layer and a westward movement of AW[6], which supports the observations reported from the coastal waters.

Decline in sea ice extent and thickness within the NEGS as well as sea ice exported from the central Arctic Ocean and melted locally can also contribute to the observed freshening. Arctic sea ice volume export through Fram Strait has been declining since the early 1990's and particularly since the early 2000's mostly due to thinning of sea ice[7]. Supplementary Fig. 6 compares the early summer (June–July–Aug) mean ice export through Fram Strait[7] to the late summer (July–Aug–Sept) sea ice volume estimate on the NEGS, assuming a one-month transit time from the Fram Strait into the NEGS. The late summer (JAS) sea ice volume in the region has reduced from approximately 265 km³ to 135 km³ from the 1990s to the 2000s while at the same time, the sea ice volume transport from the central Arctic for the summer months shows no statistically significant trend (*p*-value = 0.9; Supplementary Fig. 6). This may suggest that there is more summer sea ice melt in the NEGS region in the 2000s than in the 1990s and that local sea ice melt is likely an important source for the observed surface freshening on the NEGS. Indeed in the period 1998–2011, only the very last years showed positive amount of net sea ice melt at the surface of EGC from either local or upstream sources[38].

The contribution from winter sea ice, central Arctic Ocean liquid freshwater export (sea ice melt, precipitation, and river discharge), as well as freshwater from the GrIS, however, may outweigh the local sea ice melt in the future. The importance of these sources are expected to increase in the coming decades as mass loss of the Northeast GrIS accelerates[4] and liquid freshwater export through Fram Strait may increase[41].

AW on the shelf has warmed and shoaled significantly. These trends are consistent with the increase in ocean heat transport into the Nordic Seas since 2001[42,43] and the warming of AW entering Fram Strait since 1997 and warming of RAW which recirculates in Fram Strait before merging with the EGC and returning southward along the shelf break[44]. The increase in AW core temperature derived in our analysis (Fig. 4e) agrees with observations of temperature in the upstream WSC in Fram Strait where there is an anomalously warm period starting in 2002 and fluctuating[45] but remaining well above the long-term average with temperature anomalies of up to 2 °C in the period 2000–2018 relative to pre-2000[45]. Despite a poor understanding of the mechanisms bringing AW onto the shelf[17], the studies cited above confirm increased temperature of the AW present in the region. Trends uncovered align with reported upstream changes, implying propagation of anomalies onto the NEGS and further into selected fjords. This is further corroborated by the observation of an AW temperature increase of 0.5 °C in the period 2000–2016 relative to 1979–1999 throughout the Norske Trough, which cuts across the NEGS at ~78°N[21]. The authors further suggest that this temperature increase is reflective of warming RAW as the warming diminishes in other regions of the trough system thought to be dominated by the cooler AW branch that has circulated within the Arctic Ocean. In addition, the AW had shoaled in the 2013–2017 period compared to 1984, 1997, and 2008[22]. This is similar to the shoaling of AW reported in our study over a larger domain on the NEGS.

The stronger presence of AW has implications for the NEGS ecosystem in relation to (a) increased heat supply for ice melt, (b) change in stratification and nutrient supply, and (c) increased biological connectivity to the Atlantic (Fig. 6). Closer proximity of AW to the surface could potentially imperil the already declining sea ice extent through enhanced melting. This is particularly relevant for the southernmost reaches of the study domain, where a combination of PW thinning and density changes lead to reduced stratification (Fig. 5). A parallel can be drawn to the Barents Sea, where weakened stratification due to reduced sea ice melt water flux enabled enhanced vertical mixing, increased ocean heat content and resultantly a thinner intermediate Arctic layer[46]. Freshwater input is sufficient to counter such changes in the northern part of the near-shore NEGS, further highlighting the importance of freshwater in maintaining strong upper water column stratification—a characteristic of Arctic conditions in the NEGS. Arctic sea ice export is projected to further decline through the 21st century[41], raising the question as to whether or not Arctic liquid freshwater export and GrIS discharge will be sufficient to maintain high stratification over northern NEGS and prevent further displacement of the boundary between Arctic and Atlantic type conditions as seen over the Barents Sea[46]. However, the Arctic freshwater budget is in transition. While local sea ice melt may decrease resulting in lower freshwater contribution and greater exposure to wind-driven mixing, river discharge, and GrIS contributions are increasing and pulses in the release of freshwater from the central Arctic Ocean can contribute to maintaining stratification in the surface waters of the NEGS at irregular intervals. For deep sill fjord systems with marine terminating glaciers, warming and shoaling of AW has a further implication. Increased AW presence results in increasing heat supply to submarine glacial melting, although increased fjord temperatures do not simply translate to increased submarine melt rates, but is rather dependent on plume dynamics[47] and bathymetry[22]. Furthermore, the vertical position of the PW/AW interface is particularly important in the context of ocean heat

supply as it controls cross-sill flow of AW[15]. The shoaling of AW observed in the current study, therefore has the potential to increase heat supply to the marine terminating glaciers along NEGS. This effect has already been demonstrated for the 79N glacier, where increased heat supply in the winter of 2016/2017 was linked to thickening and shoaling of the AW layer on the NEGS[22].

During winter, cooling and wind-driven mixing erodes the surface mixed layer, replenishing surface waters with nutrients from AW below[1,48]. Ice coverage over the NEGS limits the extent of mixing and surface waters entrain nutrients from PW, which has a lower nitrate content and nitrate to phosphate ratio, than AW[38]. Over the summer months the surface mixed layer is established, ice coverage retreats, and phytoplankton become nitrate limited[1,49] (Fig. 6a). The greatest phytoplankton abundances are found at depth (so called deep chlorophyll maxima) and in particular where the photic zone crosses the AW boundary (which coincides with the nitracline). The thickness of the PW layer, and to some extent the presence of sea ice (depending on month and location), hinders mixing of nutrients from AW[1]. Despite the freshening of the surface mixed layer in recent years, which has enhanced stratification at the very surface, the energy required to mix to the 0 °C isotherm has decreased substantially (Fig. 5d). This is due to the thinning of the PW layer. In addition to potentially facilitating vertical mixing of nutrients, the PW layer thinning will also result in deepening of the photic depth due to the differences in optical properties of AW and PW[50]. PW has a higher content of colored dissolved organic matter (CDOM) resulting in greater light attenuation[50,51]. Freshening and thinning of PW will dilute and reduce the contribution of CDOM to vertical light penetration. This combined with a low contribution to light attenuation by phytoplankton in surface and PW waters due to nutrient limitation, can result in increased light penetration (photic depth). Decreased sea ice cover both increases light availability and increases energy available for vertical mixing, potentially resulting in increased summer phytoplankton productivity on the NEGS (Fig. 6b) and an increase in the abundance and productivity of higher trophic level organisms. Sea-ice-derived melt water stratification may lead to substantial changes in the biological carbon pump and cause a shift from an export to a retention system, with measurable impacts on benthic communities[52].

AW expansion and shoaling are well-documented features of change in the Eurasian Basin of the central Arctic Ocean[53] and the Barents Sea[46]. The NEGS differs by being more influenced by increased surface freshwater. Nevertheless, impacts of upstream changes in AW have been linked to the appearance of boreal biota originating from the Barents Sea including Atlantic cod, beaked redfish, and deep-sea shrimp outside known distribution ranges[54,55,56]. In the NEG fjords, the shoaling of the AW horizon sets the scene for a persistent AW presence in bottom waters where until now it was either episodic or absent (Fig. 2g). This means, that fjord and shelf ecosystems along the NEGS which have been dominated by cold Arctic water now receive water masses with positive temperatures which establishes new ecological connectivity to lower latitude ecosystems. As a consequence, the observed hydrographical changes are likely to restructure the productivity and structure of the NEGS potentially resulting in borealization of the ecosystem as observed in the Barents Sea[29,57].

## Methods
### Hydrographic data
In situ hydrographic observations from the Northeast Greenland shelf, coast and fjords were consolidated (Supplementary Table 2) from publicly available data from large oceanographic databases, including the International Council for Exploration of the Sea (ICES) oceanographic database, the World Ocean Database (WOD), the Global Temperature and Salinity Profiling Program (GTSPP), PANGAEA, and the Greenland Ecosystem Monitoring Program (GEM). Only profiles identified as "accepted value", "element appears to be correct" or

"element appears to be probably good" by the WOD and GTSPP quality controls, respectively, were downloaded. In addition, unpublished ship-based temperature and salinity profile data are included, and data from before 1935 tabulated and merged. Behrendt et al.[58] compiled many of the same publicly available data between 1980 and 2015 but here we extend the time period and provide additional data that improves data coverage, especially in the NEG fjords which are sparsely covered by UDASH.

To ensure data quality, all profiles were passed through a value range, depth inversion and gradient check. Expected value ranges were set based on literature values[59]: The temperature range was defined between −2 °C and 20 °C, salinity range from 0 to 42 and pressure range from 0 dbar to 4000 dbar. The gradient check was done to avoid erroneous increases or decreases in temperature or salinity with depth. A gradient was deemed erroneous if its absolute value exceeded 0.7 °C m$^{-1}$ for temperature and 3 units of salinity m$^{-1}$ for salinity. Data points that fell outside of the expected ranges were discarded from the analysis. Data points that were below the calculated freezing temperature of sea water were likewise removed. Finally, any duplicate profiles were identified and removed.

Prior to 1990, the majority of profiles were sampled using reversing thermometers and bottles to measure temperature and salinity at standard levels. Modern profiles were primarily sampled with CTD instrumentation, typically with an accuracy between 0.001 °C and 0.005 °C for temperature, 0.003 to 0.01 for practical salinity $S_p$ and ±2.4 dbar for pressure. In addition, Airborne Expendable CTD and few profiling floats contribute to the dataset between 2016 and 2019 with accuracy similar to that of CTD casts for profiling floats and an order of magnitude less for Airborne Expendable CTDs. In situ temperature was converted to potential temperature using the Gibbs-SeaWater (GSW) Oceanographic Toolbox built for MATLAB[60]. Subsequent analysis and computation of derived variables were done using potential temperature and practical salinity, from here on referred to as "temperature" and "salinity" unless otherwise specified.

The dataset covers the NEGS, defined as the shelf area west of the continental shelf break given by the 500 m isobath[61]. The northern extent of the study domain is set at 81.3°N north of which the EGC has different dynamics[62], and the southern extent is set at 69°N corresponding to the approximate latitude at which the EGC bifurcates and circulation dynamics take on a new character[63]. While the geographical coverage of the dataset is fair, there are seasonal and spatial biases (Supplementary Fig. 1). Profiles from between October and June have been sampled almost exclusively south of 75°N (Supplementary Fig. 1b) and represent 10.7% of the total number of profiles (4247). To avoid aliasing seasonal variability the study is therefore limited to data collected in July, August, or September. Sampling has been conducted over the entire latitudinal and temporal dimensions in these months. A total of 3791 profiles spanning almost a hundred years from 1923 to 2019 resulted from this selection.

The presence and depth of sills are particularly important characteristics albeit limited information on this topic is available. We consider a fjord to have deep or shallow sill if AW is present or absent, respectively. For the fjords in the current study, known sill depths are about 325 m and 45 m along AW inflow pathways to Dijmphna Sound and Young Sound respectively, and Scoresby Sound is 550 m at the entrance[14,22,64]. Depth estimates of fjord entrances based on the GEBCO 2020 product[65] for Godthåb Gulf, Kaiser Franz Joseph, and Kong Oscar fjords are about 110 m, 350 m, and 345 m, respectively. We highlight that these values should only be used as an estimate. In addition, strong similarity in terms of hydrographic properties and their vertical distribution between Kaiser Franz Joseph and Kong Oscar fjords allows us to regard these as one system. To minimize along fjord variability and avoid mistaking spatial for temporal variability, we focus on the main trunks of each fjord. Analysis of semivariance as a function of distance within each fjord (Supplementary note 1) showed that this

selection ensures limited spatial variability for both temperature and salinity, i.e., the difference in temperature and salinity between observations taken a given distance apart is small. This is particularly true for observations taken deeper than 30 m, whereas the surface properties showed higher degree of variability although not strictly related to space (Supplementary note 1 and Supplementary Fig. 2).

## Characterizing hydrographic change

Warm subsurface water of Atlantic origin, here simply termed Atlantic Water (AW), is characterized by temperature >0 °C and salinity >34.4. Polar Water (PW) is defined by temperature <0 °C and salinity <34.4. Hydrographic changes of PW and AW are characterized by the core temperature and salinity properties of the two water masses as well as by their vertical positioning and extent. Each metric is described in the following.

PW and AW cores are identified by the temperature minimum and maximum within the two water masses, respectively. These features are identified for each profile after linear interpolation onto a 1 m spaced grid from 0 m to 600 m. Only profiles with depths larger than 100 m were considered, thereby excluding shallow banks. In the absence of a pronounced temperature minimum, the mean properties between 40 m and 60 m were used to characterize the profile. This was particularly relevant for shallow sill fjord systems such as Young Sound. Identifying the AW core can likewise be problematic if the sampling depth does not penetrate adequately into the AW layer. To this end, temperature from the 1 m binned data of the 3 depth-levels nearest the identified temperature maximum were compared. If the temperature deviated by more than 0.01 °C between subsequent depth-levels, the core property estimates were deemed unreliable and discarded as we cannot be confident that the true AW core was sampled (following Lind et al.[66]). A final concern regarding the ability of historical data with coarse vertical resolution, to accurately estimate core properties was addressed with a small experiment. In this experiment the procedure was applied to high-resolution profiles sub-sampled at typical bottle depths (5 m, 10 m, 25 m, 50 m, 75 m, 100 m, 150 m, 200 m, 300 m, 400 m). Core properties identified using the sub-sampled and full resolution profiles were then compared. The sub-sampled profiles captured the core properties well (correlation coefficients >0.88, $p$-value < 0.05) and differences in temperature and salinity properties of both PW and AW were found to be small. However, the experiment indicated that depth of the AW core requires higher vertical resolution to be reliably estimated.

PW layer thickness is determined as the depth range with temperature below or equal to −1 °C. This definition uses −1 °C rather than 0 °C to ensure separation from AW. The upper boundary of the AW layer is defined by the 0 °C isotherm. The definition is based on temperature rather than density, as the historical data do not have the resolution necessary for reliable estimation of isopycnals. To avoid a potential near surface temperature maximum, the isotherm is searched below 40 m. Both PW layer thickness and the upper AW boundary are identified for each profile after interpolation.

## Time series construction and trend analysis

To limit across shelf discontinuity in water mass properties, a spatial constraint is adopted for time series construction. This constraint limits the spatial extent of the study region to a 100 km band from the shoreline unless the shelf is narrower in which case the 500 m isobaths is used (Fig. 1, dashed black line). Analysis of variance in the metrics described above, showed that the variance arising due to differences in sampling location is small in comparison to that arising from temporal changes (Supplementary Note 2 and Supplementary Table 1). This lends confidence to the suitability of the dataset to reflect long-term changes in water properties and time series of each of metric were constructed. Given non-random and often opportunistic sampling, simple averaging is not sufficient to provide an accurate spatial

representation[67]. Spatial sampling bias is accounted for by computing declustered statistics whereby extensively sampled areas do not dominate the time series. This is achieved by assigning each sample point a weight proportional to the area it represents, as determined by Voronoi polygons. In this approach, a polygon is drawn around each sample point in a given year fulfilling the condition that the distance from any location within a polygon is shorter to the sample point around which it has been drawn, than any other sample point. Weights are then calculated as follows:

$$w_j = \frac{A_j}{\sum_{j=1}^{n} A_j} \tag{1}$$

Where $A$ is the area of polygon $j$, and $n$ is the number of sample points. Weighting the sample points in this manner does not alter the sample value but does adjust the sample distribution, thereby avoiding giving disproportional weight to data collected closely in space. This approach is preferred over declustering with regular gridding, as issues related to defining an appropriate grid size and location of cell boundaries are avoided. Given sparse data coverage in the first half of the twentieth century, this approach can only reasonably be applied for the period after 1970. Earlier data is, however, included in the time series to provide context. The resulting statistics and final time series are thus area-weighted spatial representations that account for the heterogeneity of sparse data coverage. Linear trends are computed using least squares regression and statistical significance is evaluated with the Student's t-test. Only statistically significant trends ($p$-value < 0.05) are presented.

## Stratification criteria

Stratification is quantified with available potential energy (APE). This metric represents the work required to erode vertical density gradients and bring a given water column into a mixed state. This criterion provides information on changes occurring between the considered depths and can therefore be considered a bulk measure of stratification. APE is calculated from the 1 m binned data with units of J m$^{-3}$ [31]:

$$\text{APE} = \frac{1}{h} \int_{-z_2}^{-z_1} g\,\hat{\rho}\,z\,dz \tag{2}$$

$$\hat{\rho} = \rho(z) - \rho_{z_2} \tag{3}$$

Where $g$ is the mean gravitational acceleration in the study domain (9.83 m s$^{-2}$), $z_1$ and $z_2$ are the depths between which the metric is computed, h is the difference between $z_2$ and $z_1$, $z$ is depth measured positively upward from $z_2$, $\rho$ is potential density (kg m$^{-3}$) and $\rho_{z_2}$ is the potential density at $z_2$. APE is determined by two quantities, namely density and layer thickness. Change in APE between the 1980–2004 and 2014–2019 periods, is segmented into the relative contribution of these components (APE$'_\rho$ and APE$'_{thk}$), as well as the non-linear cross term (APE$'_c$):

$$\text{APE}' = \text{APE}'_{thk} + \text{APE}'_\rho + \text{APE}'_c \tag{4}$$

$$\text{APE}'_{thk} = \frac{1}{h} \int_{-z2}^{-z1} g\,\bar{\hat{\rho}}\,z'\,dz \tag{5}$$

$$\text{APE}'_\rho = \frac{1}{h} \int_{-z2}^{-z1} g\,\hat{\rho}\,\bar{z}\,dz \tag{6}$$

$$\text{APE}'_c = \frac{1}{h} \int_{-z2}^{-z1} g\,\hat{\rho}'\,z'\,dz \tag{7}$$

Where apostrophes denote the anomaly in the later period relative to the earlier period, and the overbar denotes the time average over the early period.

### Sea ice volume transport and NEGS sea ice volume

Monthly mean sea ice volume transport data[8], was averaged over June–July–August from 1992 to 2018. The NEGS sea ice volume is estimated from monthly mean sea ice extent (defined as the area with more than 15% sea ice concentration) in the region 70°N–82°N, 20°W–15°E from monthly sea ice concentrations at 25 × 25 km resolution from the National Snow and Ice Data Center (NSIDC, Boulder, USA) multiplied with the monthly mean effective sea ice thickness estimate across the Fram Strait between 13°W and 0°W[8] at a northern limit and the assumption that the sea ice thickness between the Fram Strait and 70°N goes linearly to zero in the summer months.

### Data availability

The hydrographic data that support the findings in this study are available from data.dtu.dk with the identifier https://doi.org/10.11583/DTU.15090207. The Fram Strait sea ice volume transport data is available via https://doi.org/10.21334/npolar.2020.696b80db. Source data are provided with this paper.

### Code availability

Code may be requested from the corresponding author.

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

## Acknowledgements

We acknowledge the considerable efforts that have gone into collecting data and making them available to the scientific community. Many of the data have been collected on expeditions financed by the Norwegian Polar Institute, Alfred Wegener Institute, Greenland Ecosystem Monitoring Programme, funded by the Danish Ministry of Climate, Energy and Utilities and the TUNU Programme. The Mineral License and Safety Authority (MLSA) and Environmental Agency for Mineral Resource Activities (EAMRA) of Greenland as part of the Joint Northeast Greenland Strategic Environmental Study Program funded sampling at NEGS in 2018. We also thank Piotr Kowalczuk and Alexey Pavlov for contributing data. C.S. acknowledges received funding from the Independent Research Fund Denmark Grant No. 9040-00266B (C.S.) and the Nordic Council of Ministers AG-Fisk Grant number (209)-2020-LEGCO (C.S.).

## Author contributions

C.G., M.S., and C.S. conceived and developed the study. C.G. assimilated and analyzed data gathered by M.S., J.C., M.G., B.K., E.M., L.S., M.W and C.G., L.S., and C.S. interpreted the results and wrote the manuscript. C.G., M.S., L.S., J.C., M.G., B.K., E.M., M.W., and C.S. clarified interpretations and commented the manuscript throughout the process.

## Competing interests

The authors declare no competing interests.
