## [Peer Review File · Nature Communications]

Warm meets fresh: vertical redistribution of principle water masses on the Northeast Greenland ShelfREVIEWER COMMENTS

Reviewer #1 (Remarks to the Author):

Review of "Warm meets fresh: Vertical redistribution of principle water masses on the Northeast Greenland Shelf" by C. Gjelstrup et al.

The manuscript provides an analysis of the multi-decadal evolution of the horizontal distribution of the vertical locations of Atlantic Water and Polar Water on the North East Greenland shelf. This is based on a compilation of hydrographic data. The calculated figures are generally insightful and worthwhile to be published. The core of the analysis shown in the paper is sound and interesting. The interpretation, however, at times goes way too far beyond what is shown in the manuscript and hence remains unconvincingly speculative.

The main conclusion is that the Atlantic Water/Polar Water interface has risen in the vertical and the authors quantify this excursion over the past half century. This has important implications for primary production, the whole ecosystem on the shelf and in fjords as well as for submarine melting of glaciers of Northeast Greenland. However, I also felt, it would helpful for the authors to refer to this recent closely related study: <https://doi.org/10.1038/s41467-021-27641-6>

I recommend major revision.

Regarding the author list and the author contributions section: It does not become obvious how some of the authors would have contributed more to this paper than some of the authors (have they been contacted?; they should not have) of the publicly available data sets that were used in this manuscript. Have those authors just not been contacted because they had provided their data freely in online data bases? And the people in the co-author list of the current manuscript "did it right" by not depositing their data, hence they were included here?

l26/27 ">60m" and "20m/decade" Why are you giving the absolute value of change in one place and the rate of change in the other place?

l30 "Increased presence of AW, including its inflow into shallow sill fjords previously void of warm AW, ..." Is this observed in your study or hypothesized as a mechanism? Should become clear from the wording used.

l38 Consider this <https://doi.org/10.1029/2021GB006961> as a reference for this region?

l42 "sea ice melt and sea ice transport"

l49 Consider this <https://doi.org/10.1002/2016JC012228> as a reference for this region?

l55 "The other half branches off westward within the Fram Strait as Recirculating Atlantic Water (RAW; 10), forming the EGC to return southwards (REF) (Fig. 1). REF: <https://doi.org/10.5194/os-14-1147-2018>

l70 Add a note along these lines: "Note that the bathymetry of many fjords is poorly known due to a dearth of observations."

l99 "Shallow sills (XXm and XXm deep) at the entrance ..."

l118 "interannual variability and changes in when in the season measurements were taken"

I139 "sill depth (XXm)"

I140 "the increase"

I165 "ubiquitously" except for the 1930 value...

I174 though → through

I199 two times: that → those

I213 A bit more discussion of the schematic might be helpful. Or at least mention that you would discuss it later.

I232 3: is that reference number three or cubed?

I239 Ref 32 is about the Norwegian Sea, arguably not the region that you are talking about in this context here.

I248 "shows no trend" Is that a statistically checked statement?

I248 "more summer sea ice melt": How big is the winter ice volume?

I252 "is driven by conditions in the remainder of the year" Are you the first one to conclude this?

I257 How about a dynamic rather than thermodynamic explanation?

I267 Consider "presence of AW" → "temperature of the AW present"

I270 and I274 Ref 19 → Ref 18

I287 "which has a deficit in nitrate relative to the AW" What do you mean by that? Obviously, the values of most nutrients are lower in PW than in AW. Deficits are typically discussed in the context of different nutrients (and e.g. their isotopes) with respect to each other.

I288 "become nitrate limited" How is this motivated from any of your calculations? Nitrate is only mentioned on this line and the immediately preceding line. Could you show a reference that nitrate ends up being the limiting nutrient rather than e.g. phosphorous, silicate?

I290 "in particular where the photic zone crosses the AW boundary" I don't think that is true on the NEGS. The AW is way too deep for light to reach that depth. Your statement might be true in the eastern Fram Strait (e.g. <https://doi.org/10.3389/fmars.2021.605225>), but you were not talking about that region.

I296 Also your Ref 37 shows that light won't reach that deep.

I299 "the productive shelf break waters south of Denmark Strait". This would need a reference.

I296-299 This statement seems way too speculative. I don't see how you have supported it with your own calculations. For a discussion of how changes in sea ice distribution may play out across the trophic chains, see e.g. <https://doi.org/10.1038/s41467-021-26943-z>

I306 "the shoaling of the AW horizon creates a tipping point effect by spilling over sills and into fjords previously dominated by PW with temperatures below -1°C" This is

where you have a convincing figure. So refer to it.

I307 "fjord and shelf ecosystems dominated by cold Arctic water" Statement seems needlessly broad.

I308 how do you know that there were no earlier spilling events in e.g. Gothab Gulf.

I312 "Close proximity" Yes, but it is still a large vertical distance away.

I283-I328 read more like an introduction chapter of a thesis than a scientific article.

I332 "is rather dependent on plume dynamics 44 and bathymetry (REF)" REF: <https://doi.org/10.1038/s41561-019-0529-x>

I335 "the potential"

I336 near → for

I363-364 <https://doi.org/10.5194/essd-10-1119-2018> did something similar also covering your region of interest. Might be worthwhile to quickly describe differences/similarities of your method.

I367 "erroneous" What do you consider erroneous in this context?

I368 Note that both vertical temperature and vertical salinity gradients can be positive and negative in this region. Maybe "if its absolute valued exceeded..."

I372 I don't think you can measure temperature with bottles. Maybe with reversing thermometers at a limited number of vertical locations that might coincide with bottles for salinity.

I374 Those numbers appear very optimistic.

I376 "with accuracy similar to that of CTD casts" Again: I think you are being too optimistic.

I382 isobaths → isobath

I382 81.3°N Why this latitude? Might be useful to mention (like you do for the southern boundary of your box) that the EGC has different dynamics at this northern location. <https://doi.org/10.5194/os-14-1147-2018>

I388 "the study is therefore limited to data collected in July, August or September" And what kind of limitations stem from that?

I395 "300m, Djimphna Sound" <https://doi.org/10.1029/2020JC017080> shows that the sill in Djimphna Sound is much shallower (<200m) than that. But <https://doi.org/10.1038/s41561-019-0529-x> show that the sill depth that matters for the 79N system (of which Djimphna Sound is a part) is 325m.

I397 "Depth estimates of fjord entrances based on the GEBCO 2020 product" And how good is this data product for these narrow undersampled fjords?

I417-419 I don't understand this! Also, if the temperature maximum is at the bottom, you could reasonably assume that the AW core temperature in surrounding shelf areas might be larger.

I425 "were found to be statistically insignificant" First, how did you define the statistical significance? Second, this cannot be, as by definition, the amplitude of an extremum

(min or max) is underestimated if you subsample a function at low resolution.

I453 equation 1 should not contain the last "n"

I460 Make doubly sure that you do not over interpret the figures keeping this comment in mind.

I463 delete "trends"

I471 I think the units should be $J m^{-2}$ not $J m^{-3}$, also Fig5a

Equation 3 might be easier if you replace ρ_0 by $\rho(z_2)$, also I474

I472 Should h not be $h=z_2-z_1$?

I475 Space after z_2 . "water column thickness": that cannot have changed between the those periods (tides, geologic changes). I think you meant "layer thickness".

Equation 5: "z' " How can the integration variable become an anomaly? That is not how integral notation works! I think this decomposition is meaningless, or at least not sufficiently motivated here.

I482 "Jun/Jul/Aug" Why not "Jul/Aug/Sep", compare I388?

I483 The DOI should go to the data availability section below.

I494 I was not able to locate this on data.dtu.dk or via <https://doi.org> Maybe the link is not active (yet)?

I603 Year missing.

I630 Consider "data gathered by M.S., J.C., M. G., B.K., E.M., L.S., M.W., C.S. and others (see Table S2)". See my comment on the author contributions above.

Fig1 The light shading is confusing. It is too light to be clearly visible. Also, it adds to the background bathymetry (which is also missing a colorbar!) to result in apparent depth artifacts such as at $73.5^\circ N$ where the figure makes it look like the shelf gets much shallower north of a straight east-west line. Consider black hatching rather than shading.

Fig1 Why do you not show bathymetry in the fjords instead of the true color images? Probably because it is not of good enough quality to be meaningful?

Fig2 Consistent y-axis across all panels from 0 to 500m (and not e.g. 0-200m as in second row) to reduce the information that the reader needs to take in before being able to compare the different panels.

The yellow (on white) lines are barely visible and the yellow numbers indicating the number of profiles is entirely non-legible. The fact that the colorbar is non-uniformly spaced (entire decades are missing, but after 2002 every year features) is a very poor choice. If you decide to keep it, at least clearly state this in the figure caption.

Fig2 caption: "Mean July/August/September profiles of ..."

Fig3 Again the non-uniform spacing of the x-axis (like the colorbar in Fig2) is a very poor choice.

The font used for the x-axis labels does not work (in panel b the second value: is it 1932 or 1982?).

Could you not also show the location of the temperature maximum (AW core), at least for some of the profiles?

Fig3 caption "... for the three deep sill fjord systems during July/August/September". This is important information that should not be buried so deeply in the method section. Rather repeat it in every figure caption.

Fig4 What are the box/whiskers? Min/max or mean+-standard deviation or XX-percentiles? Also in Fig5, it is not defined what is shown.

Fig5 I have never heard of an "integrated density deviation" before. I think presenting an "averaged density deviation" in kg m^{-3} would be more effective.

Fig6 Why is the red shading on the left lighter than on the right? What information do you intend to convey by that? Explain.

It appears that you have drawn a sill to a fjord on the far left of the panels. Mention that this is what you are showing rather than a coastline or shelfbreak.

What is the vertical red line on the far right of panel c over the green?

Fig6 caption: Do you have any idea how deep the light penetration is (<http://dx.doi.org/10.1016/j.jmarsys.2014.11.001> Fig7 states 40m in the WSC; others shallower)?

Are you not getting a bit ahead of yourself with the schematic?

Have you shown/discussed nutrient data to draw these conclusions?

Have you motivated what that "green layer" is? Where did it come from? How did it get established?

What do you know/show in the paper about the benthos?

Are your schematic particles on the sea floor exported phytoplankton or benthic organisms that consumed exported material? Compare schematic in

<https://doi.org/10.1038/s41467-021-26943-z> and how every aspect of that schematic is backed up with data in the paper.

FigS1b legend: In addition to the ones at 1 and 76, why do you show an in-between sized marker without an attached number?

I66 "The number of profiles" Is the scaling according to the radius or the area of the circle?

FigS2 Fontsize way too small! Non-uniformly spaced colorbar.

FigS4 Add a horizontal line along 0. This would make it visually much easier to judge the size (and sign) of the transport.

I175/182/186 The REFs 12/14/15 that you give here are data averaged on bottle locations. You had intended to cite these: <https://doi.org/10.1594/PANGAEA.863063> <https://doi.org/10.1594/PANGAEA.871025> <https://doi.org/10.1594/PANGAEA.885358>

Reviewer #2 (Remarks to the Author):

Review of "Warm meets fresh: Vertical redistribution of principle water masses on the Northeast Greenland Shelf" by Gjelstrup et al.

In this manuscript, the authors describe changes in the vertical water column structure on the Northeast Greenland Shelf over the past ~90 years. Historical profiles from various sources were compiled into a single database, the data were quality-controlled,

and then analyzed. In large part, the authors focus on linear trends, albeit over different time periods for each region, though interannual variability at some sites was also mentioned.

I found the subject material interesting and the writing clear, but I remain unconvinced of the paper's central results. In particular, there was little to no mention of spatial variability within the regions of interest, especially as it pertains to the limited coverage of the observations. On one hand the authors did a fine job of only selecting summer data so as to not contaminate their signals with seasonality, but the same care was not taken (or was not obvious to the reader) to control for spatial variability. For example, if some of these vertical profiles were taken near the heads of fjords in the summer months, we may expect to see a very strong surface fresh water layer. And near the fjord mouth, we would expect to see more influence in the bottom layers from the warm and saline Atlantic-sourced waters. If the distribution of the observations changes through time, then this spatial variability could appear as temporal variability in their analysis and refute their result that these are secular changes in the fjord properties. This point holds even when there are multiple profiles averaged together in a given year because a single profile with a strong property signal can dominate many profiles with weak property signals (i.e., the underlying distribution is not normal, and the mean may not be representative of any individual profile). So where these profiles were taken is incredibly important, but not explained in the paper. One way to account for this in a qualitative way is to show a map of profile locations for each year in each fjord/region. A more quantitative method could be to construct a mean spatial map for each fjord/region by gridding all the available data, and then subtracting that mean field from the individual profiles, and thereby constructing property anomalies from the mean. This would require there be sufficient sampling through all years to construct a reliable mean fjord structure, which I am not sure is the case. I encourage the authors to think of other methods to account for the spatial variability.

I also found that the introduction section did not follow a logical sequence toward a motivation for the study, and instead consisted of a list of somewhat redundant list of information about why the Northeast Greenland Shelf is important. In this section, I found the writing to be clear from a grammatical point of view, but it wasn't always clear why one sentence/thought followed the other. A simple motivation would be that we really know very little about these regions due to their ice cover and inhospitable weather, and yet there is this great historical database of profile data that have not been compared to more modern measurements.

Some of the figures were also hard to interpret, especially Fig. 2. My understanding of Fig. 2 is that for each vertical profile shown in each panel, there are some number (n) of vertical profiles averaged over that year. The n for each year (color-coded by year) is shown above the plot. From the text and captions, it appears the goal of the plot is to show the time variability of these profiles, but this arrangement of the data (and especially the use of yellow in the colormap) does not make the time variability readily apparent - it is hard to visualize time variability when profile data overlap one another so much. Linear trends are apparent, but any sort of variability cannot be identified. In addition, it is not apparent that the average profile within each year accurately summarizes the conditions of the fjord/region in that year. It would instructive to see how large the spread is within each year, to compare that to the trend the authors discuss.

I found the discussion of APE to be strange, partly because I don't see how APE can be calculated from a one-dimensional profile - doesn't APE requires a sloping of isopycnals between two or more spatial locations? On the other hand, I don't know what APE added to the discussion in the paper. Could a simple measure of stratification make the same point and be more clear to the reader?

I was also a bit underwhelmed by the discussion of mechanisms. If there is more Atlantic Water penetrating into these fjords, what is causing that to occur? Are the

winds different now than they were in the 1930s? Or maybe less sea ice cover makes the winds more 'effective' at moving the water column?

In Fig. 6 and the accompanying text, I was thoroughly confused by why the authors started discussing phytoplankton seasonal cycles. Nothing prior to this point in the text would indicate that the authors were interested in phytoplankton. Maybe this is where a more complete introduction with a motivation for the study would help the reader – could the authors mention in the introduction how this study will explain phytoplankton seasonality?

In closing, I want to reassure the authors that I do believe a study like this is warranted and I commend the authors for the amount of time and effort that compiling these data must have taken. I also think there is an interesting signal in these data, but the authors need to do a bit more to make their argument more convincing. Maybe my point about the spatial variability is not valid and the sampling distribution IS sufficient, but the authors need to demonstrate that convincingly so that their hard work will be rewarded by the community.

Reply to review

Warm meets fresh: Atlantification of Northeast Greenland Shelf and Fjords

Thank you for the thorough and constructive reviews of our manuscript. We have revised the manuscript according to the recommendations raised by the reviewers addressing all points.

Below you will find a point-by-point reply to the specific points raised by the reviewers. The reviewer comments are highlighted in grey and our responses are below.

Specific comments from Reviewer 1:

The core of the analysis shown in the paper is sound and interesting. The interpretation, however, at times goes way too far beyond what is shown in the manuscript and hence remains unconvincingly speculative.

The reviewer regards our interpretation of the findings as sound but questions our speculation on the implications of the physical oceanographic changes reported. In our revised manuscript, we have clarified which interpretations are directly supported by our analysis and which statements are intended to put our findings in perspective. We hope the revised manuscript reads clearer and the reader has confidence in the paper.

The main conclusion is that the Atlantic Water/Polar Water interface has risen in the vertical and the authors quantify this excursion over the past half century. This has important implications for primary production, the whole ecosystem on the shelf and in fjords as well as for submarine melting of glaciers of Northeast Greenland. However, I also felt, it would be helpful for the authors to refer to this recent closely related study: <https://doi.org/10.1038/s41467-021-27641-6>

Having been published while our manuscript was in review, we were not aware of this publication. We have now incorporated the findings into our manuscript (See introduction). As the Moore et al study is focused on the shelf break EGC it is not highly relevant to our discussion of the changes we report for the coastal waters on the shelf itself.

Regarding the author list and the author contributions section: It does not become obvious how some of the authors would have contributed more to this paper than some of the authors (have they been contacted?; they should not have) of the publicly available data sets that were used in this manuscript. Have those authors just not been contacted because they had provided their data freely in online data bases? And the people in the co-author list of the current manuscript “did it right” by not depositing their data, hence they were included here? A considerable amount of time was spent gathering, quality assessing and organizing data that was, until now, not publicly available. No authors were contacted because they had “hidden” data, but rather because they had appropriate data, and they could contribute with constructive input and scientific analysis. The paper is a result of data mining at the institutions involved, to try to make publicly available as much as possible. In addition to what is stated in author contributions, we can add the following explanation. The initiative for this paper arose from several parallel discussions: 1) Initial comparisons for Scoresby Sound based on Polarstern and Maria S. Merian cruises (B. Koch & C. Stedmon); 2) Discussions in the ICES working group for the East Greenland Shelf (<https://www.ices.dk/community/groups/Pages/WGIEAGS.aspx>) where we have tried to gather baseline data on how the physical environment has changed in the region so that we can link later to ecological changes, including publicly available and previously unpublished data (WG participants, C. Stedmon, C. Gjelstrup, M. Sejr, E. Friis Møller, Mie Winding, J. Christiansen); 3) an effort to digitalize data from early research carried out before the 1950s (M. Sejr); and 4) long term data collection and analysis effort in the Fram Strait (extending to Dømmphna Sound) and North East Greenland fjords (L. de Steur, M. Granskog, M. Sejr, M. Winding, J. Christiansen, C. Stedmon).

126/27 “>60m” and “20m/decade” Why are you giving the absolute value of change in one place and the rate of change in the other place?

Changed from “20m/decade” to “> 50m” for consistency.

l30 "Increased presence of AW, including its inflow into shallow sill fjords previously void of warm AW, ..." Is this observed in your study or hypothesized as a mechanism? Should become clear from the wording used. Now clarified. (See results for Godthåbs Gulf).

l38 Consider this <https://doi.org/10.1029/2021GB006961> as a reference for this region?
Reference added.

l42 "sea ice melt and sea ice transport"
The sentence has been clarified.

l49 Consider this <https://doi.org/10.1002/2016JC012228> as a reference for this region?
Reference to Aagaard and Coachman 1968 has been added in addition to the suggested reference to reflect more original work in the region and as the EGC was not discovered as late as 2017.

l55 "The other half branches off westward within the Fram Strait as Recirculating Atlantic Water (RAW; 10), forming the EGC to return southwards (REF) (Fig. 1). REF: <https://doi.org/10.5194/os-14-1147-2018>
Reference to Rudels et al., 2002 has been added.

l70 Add a note along these lines: "Note that the bathymetry of many fjords is poorly known due to a dearth of observations."
Comment added, now on lines 70-71.

l99 "Shallow sills (XXm and XXm deep) at the entrance ..."
Sill depths have been added.

l118 "interannual variability and changes in when in the season measurements were taken"
Corrected.

l139 "sill depth (XXm)"
Sill depth has been added.

l140 "the increase"
Corrected.

l165 "ubiquitously" except for the 1930 value...
Here, "ubiquitously" is in reference to the past four decades, i.e. 1980-2020.

l174 though → through
Corrected.

l199 two times: that → those
Corrected.

l213 A bit more discussion of the schematic might be helpful. Or at least mention that you would discuss it later.
The sentence has been edited to encompass the suggestion (now lines 226-228).

l232 3: is that reference number three or cubed?
Cubed. The white space between "km" and "3" has been removed for clarity.

l239 Ref 32 is about the Norwegian Sea, arguably not the region that you are talking about in this context here.
Reference 32 is indeed about the Norwegian Sea, which is not appropriate in this context. The correct reference is Karpouzoglou et al., 2022 (<https://agupubs.onlinelibrary.wiley.com/doi/10.1029/2021JC018122>) and is now included instead. Thank you for spotting this mistake.

l248 "shows no trend" Is that a statistically checked statement?
The trend in summer (JJA) sea ice transport volume through Fram Strait is statistically insignificant with p-value = 0.47. The text has been edited to clarify this distinction.

I248 “more summer sea ice melt”: How big is the winter ice volume?

We did not calculate this, since we want to identify the changes in summer only as the hydrography is restricted to summer.

I252 “is driven by conditions in the remainder of the year” Are you the first one to conclude this?

We removed this sentence since we do not show this explicitly here as a result and it is not relevant to the scope of the paper.

I257 How about a dynamic rather than thermodynamic explanation?

This is what we are insinuating in the sentences in this section.

” The contribution from central Arctic Ocean liquid freshwater export (sea ice melt, precipitation, and river discharge), as well as freshwater from the GrIS, however, may outweigh the local sea ice melt in the future. The importance of these sources are expected to increase in the coming decades as mass loss of the Northeast GrIS accelerates³ and liquid freshwater export through Fram Strait may increase³³.”

I267 Consider “presence of AW” → “temperature of the AW present”

Thank you for the suggestion, the edit has been made.

I270 and I274 Ref 19 → Ref 18

Corrected.

I287 “which has a deficit in nitrate relative to the AW” What do you mean by that? Obviously, the values of most nutrients are lower in PW than in AW. Deficits are typically discussed in the context of different nutrients (and e.g. their isotopes) with respect to each other.

This has been clarified. We are referring to lower nitrate concentrations but also lower nitrate to phosphate ratios (i.e. a nitrate deficit).

I288 “become nitrate limited” How is this motivated from any of your calculations? Nitrate is only mentioned on this line and the immediately preceding line. Could you show a reference that nitrate ends up being the limiting nutrient rather than e.g. phosphorous, silicate?

This is now clarified with a reference. PW has a lower N:P ratio due to denitrification in the central Arctic.

I290 “in particular where the photic zone crosses the AW boundary” I don’t think that is true on the NEGS. The AW is way too deep for light to reach that depth. Your statement might be true in the eastern Fram Strait (e.g. <https://doi.org/10.3389/fmars.2021.605225>), but you were not talking about that region.

We have edited our figure. Agree the photic zone does not extend that deep. The photic zone extends to about 50m in eastern Fram Strait (vertical structure with only a mixed layer and AW below). Our earlier work shows that on the Western side it does not currently extend deeper than 35 m or so (Pavlov et al 2015). It is essentially following the base of the mixed layer and is now adjusted in the figure. Explanation is also modified.

Should also be noted that a large model intercomparison could not resolve which factor, light or nutrient availability, is the most important limiting factor (Popova et al 2012, doi:10.1029/2011JC007112.).

I296 Also your Ref 37 shows that light won’t reach that deep.

Correct. See above.

I299 “the productive shelf break waters south of Denmark Strait”. This would need a reference.

Now added.

I296-299 This statement seems way too speculative. I don’t see how you have supported it with your own calculations. For a discussion of how changes in sea ice distribution may play out across the trophic chains, see e.g. <https://doi.org/10.1038/s41467-021-26943-z>

We have rephrased and expanded this part of the manuscript. This should now make it clearer to the reader that we are attempting to put our findings into perspective and speculate on potential impact that these changes can have. The paper provided by the reviewer is indeed interesting and we now cite it in the manuscript. Having been published weeks before we submitted this manuscript to review, we had not time to consider it in our original submission. However, we do note that, it is focused on the AW side of the region,

mainly the deep parts of the Fram Strait and is not directly applicable to the NEGS. What we are trying to sketch out is the situation on the shelf where PW is persistently present above AW. However, their figure 8 is much inspiration for our new version.

l306 “the shoaling of the AW horizon creates a tipping point effect by spilling over sills and into fjords previously dominated by PW with temperatures below -1°C ” This is where you have a convincing figure. So refer to it. Done.

l307 “fjord and shelf ecosystems dominated by cold Arctic water” Statement seems needlessly broad. We have made more specific (East coast).

l308 how do you know that there were no earlier spilling events in e.g. Gothab Gulf. This cannot be ruled out completely, however with the shoaling of the boundary between PW and AW above the sill this result in a persistent AW presence. Has now been rephrased.

l312 “Close proximity” Yes, but it is still a large vertical distance away. Please note the sentence directly after this where we indicate downstream effects.

l283-l328 read more like an introduction chapter of a thesis than a scientific article. Unclear how to address this point as it offers little insight. This section has been expanded and rephrased and hopefully improved in readability.

l332 “is rather dependent on plume dynamics 44 and bathymetry (REF)” REF: <https://doi.org/10.1038/s41561-019-0529-x> Reference has been included.

l335 “the potential” Corrected.

l336 near → for Corrected.

l363-364 <https://doi.org/10.5194/essd-10-1119-2018> did something similar also covering your region of interest. Might be worthwhile to quickly describe differences/similarities of your method. The paper has now been included in the methods section (see “Hydrographic data”) alongside a brief explanation of what our dataset provides in addition to the UDASH compilation.

l367 “erroneous” What do you consider erroneous in this context? Here, “erroneous” refers to a gradient with absolute value exceeding $0.7^{\circ}\text{C m}^{-1}$ for temperature or 3 units of salinity per. meter for salinity as specified in the following sentence. “unreliable” in the following sentence (l368) has been edited to erroneous for clarification.

l368 Note that both vertical temperature and vertical salinity gradients can be positive and negative in this region. Maybe “if its absolute valued exceeded...” The suggestion has been implemented.

l372 I don’t think you can measure temperature with bottles. Maybe with reversing thermometers at a limited number of vertical locations that might coincide with bottles for salinity. This clarification has been included in the text.

l374 Those numbers appear very optimistic. The cited accuracies are based on documentation from the various data sources used to compile the dataset. E.g. the World Ocean Database reports typical CTD accuracies between ± 0.001 to $\pm 0.005^{\circ}\text{C}$ for temperature and ± 0.003 to ± 0.02 for salinity (please see: https://www.ncei.noaa.gov/sites/default/files/2020-04/wod_intro_0.pdf).

1376 “with accuracy similar to that of CTD casts” Again: I think you are being too optimistic.

Data collected with profiling floats stem from the ARGO programme, which reports accuracies of ± 0.002 °C and ± 0.01 for temperature and salinity respectively (please see: https://argo.ucsd.edu/data/data-faq/#:~:text=the%20PSAL_ADJUSTED%20variable,-,How%20accurate%20is%20the%20Argo%20data%3F,sometimes%20affected%20by%20sensor%20drift). Data collected with Airborne Expendable CTD stem from NASA’s Oceans Melting Greenland programme, with reported accuracy of ± 0.035 for temperature and ± 0.05 for salinity (e.g. chrome-extension://efaidnbmnnnibpcajpcglclefindmkaj/https://www.lockheedmartin.com/content/dam/lockheed-martin/rms/documents/oceanographic-instrumentation/Marion_MK21_20050051.pdf). This is admittedly worse than the cited accuracy and the text has been edited accordingly.

1382 isobaths → isobath

Corrected.

1388 81.3°N Why this latitude? Might be useful to mention (like you do for the southern boundary of your box) that the EGC has different dynamics at this northern location. <https://doi.org/10.5194/os-14-1147-2018>

A note has been added along with the suggested reference.

1388 “the study is therefore limited to data collected in July, August or September” And what kind of limitations stem from that?

The greatest impact this would have is for the surface waters and we strive to mention this when we refer to changes in the summer surface water properties. We do not know deeper features, such as the AW PW boundary to exhibit substantial seasonality.

1395 “300m, Djimphna Sound” <https://doi.org/10.1029/2020JC017080> shows that the sill in Djimphna Sound is much shallower (<200m) than that. But <https://doi.org/10.1038/s41561-019-0529-x> show that the sill depth that matters for the 79N system (of which Djimphna Sound is a part) is 325m.

Corrected.

1397 “Depth estimates of fjord entrances based on the GEBCO 2020 product” And how good is this data product for these narrow undersampled fjords?

The horizontal resolution of the GEBCO product is 15 arc seconds or 450 m. The product suppliers state the following regarding product accuracy: “While every effort has been made to ensure reliability within the limits of present knowledge, the accuracy and completeness of The GEBCO Grid cannot be guaranteed” (https://www.gebco.net/data_and_products/gridded_bathymetry_data/gebco_2021/). This reflects the lack of bathymetric measurements on the NEGS and the cited depths should only be considered estimates. A note underscoring this fact has been added to the text.

1417-1419 I don’t understand this! Also, if the temperature maximum is at the bottom, you could reasonably assume that the AW core temperature in surrounding shelf areas might be larger.

This exercise was done to guard against underestimating AW core temperatures due to insufficient sampling depth, which as you point out, could occur if the temperature maximum is e.g. at the bottom. We compared the identified AW core temperature with the temperature at the depth levels immediately surrounding the AW core to assess whether or not the temperature had stabilised. If the temperature of the surrounding depth layers was very similar to that at the identified AW core, the estimated AW core temperature was accepted. If on the other hand, the temperature of the surrounding depth levels was very different from that at the identified AW core, we cannot be confident that the true AW core was sampled, as there is indication that temperature continues to increase with depth. In this case we discarded the AW core estimate. The text has been altered to clarify the above.

1425 “were found to be statistically insignificant” First, how did you define the statistical significance? Second, this cannot be, as by definition, the amplitude of an extremum (min or max) is underestimated if you subsample a function at low resolution.

The ability of the subsampled profiles to capture important hydrographic features within a given profile was evaluated using the correlation coefficient between metrics derived from full resolution profiles and those

derived from the subsampled profiles. The correlation was considered statistically significant when the associated p-value was equal to or less than 0.05. The text has been edited to make this distinction clear. Regarding your second point: the vertical position of PW and AW cores can be either over or underestimated from the subsampled profiles. If the core depth is underestimated, the associated salinity of the PW core or AW core is typically also underestimated and vice versa if the core depth is overestimated. It is therefore possible to overestimate the depth and salinity metrics from the subsampled profiles, but the temperature metrics will always be underestimated as you point out.

l453 equation 1 should not contain the last “n”

Corrected.

l460 Make doubly sure that you do not over interpret the figures keeping this comment in mind.

Agreed.

l463 delete “trends”

Corrected.

l471 I think the units should be $J m^{-2}$ not $J m^{-3}$, also Fig5a

Please note that the integrand in equation 2 is scaled by $1/h$, where h is $z_2 - z_1$ in units of meters. Thus, the units of APE are Joules per unit volume rather than Joules per unit area.

Equation 3 might be easier if you replace ρ_0 by $\rho(z_2)$, also l474

Corrected.

l472 Should h not be $h = z_2 - z_1$?

That is correct, h is the thickness of the layer between z_2 and z_1 . The text has been clarified.

l475 Space after z_2 . “water column thickness”: that cannot have changed between the those periods (tides, geologic changes). I think you meant “layer thickness”.

Corrected.

Equation 5: “ z ” How can the integration variable become an anomaly? That is not how integral notation works! I think this decomposition is meaningless, or at least not sufficiently motivated here.

The decomposition helps us interpret the observed changes in APE and makes it possible to allocate these changes to the part driven by the observed surface and PW freshening (density term) and the part due to the observed PW thinning/upwards movement of the $0^\circ C$ isotherm (thickness term). Together with the overall change in APE this enables us to identify dominant drivers of change and discuss targeted ecosystem effects. We have rephrased the results section on APE to clarify why it is a convenient measure of stratification and now use the results of the decomposition more actively in our discussion.

l482 “Jun/Jul/Aug” Why not “Jul/Aug/Sep”, compare l388?

We use JJA rather than JAS to derived sea ice volume transport estimates through Fram Strait as we assume a one-month transit time from Fram Strait into the NEGS.

l483 The DOI should go to the data availability section below.

Corrected.

l494 I was not able to locate this on data.dtu.dk or via <https://doi.org> Maybe the link is not active (yet)?

The link will be made active upon publication.

l603 Year missing.

Corrected.

l630 Consider “data gathered by M.S., J.C., M. G., B.K., E.M., L.S., M.W., C.S. and others (see Table S2)”. See my comment on the author contributions above.

We have responded above.

Fig1 The light shading is confusing. It is too light to be clearly visible. Also, it adds to the background bathymetry (which is also missing a colorbar!) to result in apparent depth artifacts such as at 73.5°N where the figure makes it look like the shelf gets much shallower north of a straight east-west line. Consider black hatching rather than shading.

Figure 1 has been updated according to the above suggestions.

Fig1 Why do you not show bathymetry in the fjords instead of the true color images? Probably because it is not of good enough quality to be meaningful?

The resolution of bathymetry in the fjords is, as you mention, poor. We therefore prefer the true colour images to provide a reference for the fjords.

Fig2 Consistent y-axis across all panels from 0 to 500m (and not e.g. 0-200m as in second row) to reduce the information that the reader needs to take in before being able to compare the different panels.

The y-axes on Figure 2 have been made consistent across all panels as suggested.

The yellow (on white) lines are barely visible and the yellow numbers indicating the number of profiles is entirely non-legible. The fact that the colorbar is non-uniformly spaced (entire decades are missing, but after 2002 every year features) is a very poor choice. If you decide to keep it, at least clearly state this in the figure caption.

Figure 2 has been updated with a new linear colour scale as suggested to ease interpretation.

Fig2 caption: "Mean July/August/September profiles of ..."

Corrected.

Fig3 Again the non-uniform spacing of the x-axis (like the colorbar in Fig2) is a very poor choice.

The non-linear x-axis was chosen to minimise empty space on the figure. A note to make the reader aware of the non-linear x-axis has been included in the figure caption.

The font used for the x-axis labels does not work (in panel b the second value: is it 1932 or 1982?).

Font has been changed to make labels more legible.

Could you not also show the location of the temperature maximum (AW core), at least for some of the profiles?

The depth of the AW core has been included for years in which a temperature maximum could be identified reliably, i.e. there was a distinctive temperature maximum. Figure caption has also been updated to reflect this change.

Fig3 caption "... for the three deep sill fjord systems during July/August/September". This is important information that should not be buried so deeply in the method section. Rather repeat it in every figure caption.

This specification has been included in the captions for all calculated figures.

Fig4 What are the box/whiskers? Min/max or mean±standard deviation or XX-percentiles? Also in Fig5, it is not defined what is shown.

Error bars and box whiskers on figures 4 and 5 have been defined in the respective figure captions.

Fig5 I have never heard of an "integrated density deviation" before. I think presenting an "averaged density deviation" in kg m^{-3} would be more effective.

Figure 5c has been updated to show the mean density deviation as suggested. The figure caption has been updated accordingly.

Fig6 Why is the red shading on the left lighter than on the right? What information do you intend to convey by that? Explain.

This is now corrected. No need.

It appears that you have drawn a sill to a fjord on the far left of the panels. Mention that this is what you are showing rather than a coastline or shelfbreak.

Corrected.

What is the vertical red line on the far right of panel c over the green?
This is an error (it is the layer behind in the drawing). Now corrected.

Fig6 caption: Do you have any idea how deep the light penetration is (<http://dx.doi.org/10.1016/j.jmarsys.2014.11.001> Fig7 states 40m in the WSC; others shallower)?
Are you not getting a bit ahead of yourself with the schematic?
Have you shown/discussed nutrient data to draw these conclusions?
Have you motivated what that “green layer” is? Where did it come from? How did it get established?
What do you know/show in the paper about the benthos?
Are your schematic particles on the sea floor exported phytoplankton or benthic organisms that consumed exported material? Compare schematic in <https://doi.org/10.1038/s41467-021-26943-z> and how every aspect of that schematic is backed up with data in the paper.

We have attempted to improve this version with inspiration from the Von Appen et al. paper. It should be noted however that the goal with the figure is to put the processes/conditions discussed into perspective. What does it matter that the PW layer is thinning and it is has become easier to mix down to AW? Here we speculate on the potential impacts and provide a schematic that summarises these impacts. So, it draws on information from many studies and we have included appropriate references.

FigS1b legend: In addition to the ones at 1 and 76, why do you show an in-between sized marker without an attached number?

The in-between sized marker has been removed for clarity.

l66 “The number of profiles” Is the scaling according to the radius or the area of the circle?
Scaling is according to the area of the circle. This has been clarified in the figure caption.

FigS2 Fontsize way too small! Non-uniformly spaced colorbar.

Figure S2 has been edited according to both points.

FigS4 Add a horizontal line along 0. This would make it visually much easier to judge the size (and sign) of the transport.

Done.

l175/182/186 The REFs 12/14/15 that you give here are data averaged on bottle locations. You had intended to cite these: <https://doi.org/10.1594/PANGAEA.863063> <https://doi.org/10.1594/PANGAEA.871025> <https://doi.org/10.1594/PANGAEA.885358>

Thank you for spotting this. The reference list has been updated accordingly.

Specific comments from Reviewer 2:

I found the subject material interesting and the writing clear, but I remain unconvinced of the paper’s central results. In particular, there was little to no mention of spatial variability within the regions of interest, especially as it pertains to the limited coverage of the observations. On one hand the authors did a fine job of only selecting summer data so as to not contaminate their signals with seasonality, but the same care was not taken (or was not obvious to the reader) to control for spatial variability. For example, if some of these vertical profiles were taken near the heads of fjords in the summer months, we may expect to see a very strong surface fresh water layer. And near the fjord mouth, we would expect to see more influence in the bottom layers from the warm and saline Atlantic-sourced waters. If the distribution of the observations changes through time, then this spatial variability could appear as temporal variability in their analysis and refute their result that these are secular changes in the fjord properties. This point holds even when there are multiple profiles averaged together in a given year because a single profile with a strong property signal can dominate many profiles with weak property signals (i.e., the underlying distribution is not normal, and the mean may not be representative of any individual profile). So where these profiles were taken is incredibly important, but not explained in the

paper. One way to account for this in a qualitative way is to show a map of profile locations for each year in each fjord/region. A more quantitative method could be to construct a mean spatial map for each fjord/region by gridding all the available data, and then subtracting that mean field from the individual profiles, and thereby constructing property anomalies from the mean. This would require there be sufficient sampling through all years to construct a reliable mean fjord structure, which I am not sure is the case. I encourage the authors to think of other methods to account for the spatial variability.

The reviewer suggests that the findings may be confounded with spatial variability due to the sampling location within fjords, which differs between years. The near-surface does not matter (Seasonal mixed layer) for the core findings of this paper, because the PW/AW boundary is at depth and would not be affected if the surface meltwaters in summer are well stratified. However, to address the concern, we have now included a formal analysis of spatial variability of water properties within each fjord. This analysis is included in the supplementary information alongside maps of the sampling location for each year as suggested by the reviewer. The methods section has also been updated to clarify how we have accounted for spatial variability.

I also found that the introduction section did not follow a logical sequence toward a motivation for the study, and instead consisted of a list of somewhat redundant list of information about why the Northeast Greenland Shelf is important. In this section, I found the writing to be clear from a grammatical point of view, but it wasn't always clear why one sentence/thought followed the other. A simple motivation would be that we really know very little about these regions due to their ice cover and inhospitable weather, and yet there is this great historical database of profile data that have not been compared to more modern measurements.

In the revised the Introduction we attempt to address the reviewers points by restructuring and rephrasing the text to guide the reader towards the motivation for our study. Furthermore, we now include an expanded section on ecosystem considerations to align the introduction with the discussion later on. We hope these alterations improve the overall readability of the manuscript.

Some of the figures were also hard to interpret, especially Fig. 2. My understanding of Fig. 2 is that for each vertical profile shown in each panel, there are some number (n) of vertical profiles averaged over that year. The n for each year (color-coded by year) is shown above the plot. From the text and captions, it appears the goal of the plot is to show the time variability of these profiles, but this arrangement of the data (and especially the use of yellow in the colormap) does not make the time variability readily apparent - it is hard to visualize time variability when profile data overlap one another so much. Linear trends are apparent, but any sort of variability cannot be identified. In addition, it is not apparent that the average profile within each year accurately summarizes the conditions of the fjord/region in that year. It would instructive to see how large the spread is within each year, to compare that to the trend the authors discuss.

We have edited Fig. 2 by changing the colourmap and using consistent scaling of the x and y-axis of the profile plots and T/S diagrams. We hope this improves the interpretability of the figure. In order to better visualise the trends over time the reader is directed to Figure 4. Here the standard error of the means are also shown which gives an indication of the variability. We have added an additional figure to the sup info which includes T-S figures of the available data.

I found the discussion of APE to be strange, partly because I don't see how APE can be calculated from a one-dimensional profile – doesn't APE requires a sloping of isopycnals between two or more spatial locations? On the other hand, I don't know what APE added to the discussion in the paper. Could a simple measure of stratification make the same point and be more clear to the reader?

APE can be (and is) calculated from a vertical density profile and is a robust measure of stratification, e.g. see <https://doi.org/10.1088/1748-9326/aaec1e> or <https://doi.org/10.1007/s00382-019-04816-y>. In our study, we are interested in the resistance to mixing exerted by the surface and PW layers isolating heat and nutrients from the AW below. Using an even simpler measure of stratification such as the density difference between two depth levels would obscure this objective, as changes occurring between the two chosen depths are not reflected. For example, in the northern region considered in our study, we find an increased mean density difference between the 1980-2004 and 2004-2019 periods. All else being equal this would lead to greater stratification as reported in <https://doi.org/10.1029/2021GB006961>. But by accounting for the reduced PW layer thickness (upwards movement of the 0 °C isotherm) we conversely show that there has been a decrease in the resistance to mixing between the two periods (Fig. 5a-c). APE thus accounts for the integrated effects of hydrographic change and is therefore the more appropriate metric for our objective and measure of stratification in general.

I was also a bit underwhelmed by the discussion of mechanisms. If there is more Atlantic Water penetrating into these fjords, what is causing that to occur? Are the winds different now than they were in the 1930s? Or maybe less sea ice cover makes the winds more 'effective' at moving the water column?

This is beyond the current scope of the manuscript and requires further investigation. Our focus here is to gather available data to confirm or reject anecdotal evidence arising from many years of sampling in the region. This arose from the ICES WG discussion. We should also keep in mind that the manuscript is already at the maximum permissible length for the journal. We have now started a broader scale analysis of change in the region, incorporating satellite data products and hope to identify key mechanisms in a future study.

In Fig. 6 and the accompanying text, I was thoroughly confused by why the authors started discussing phytoplankton seasonal cycles. Nothing prior to this point in the text would indicate that the authors were interested in phytoplankton. Maybe this is where a more complete introduction with a motivation for the study would help the reader – could the authors mention in the introduction how this study will explain phytoplankton seasonality?

In the revised manuscript, we have added a paragraph in the introduction, outlining some potential impacts that changes in hydrography may have on the ecosystem. This is then further elaborated in the discussion of Figure 6 where the potential impacts of our findings on the ecosystem are evaluated. E.g. What are the implications of PW layer thinning and reduced resistance to mixing down to the AW?

In closing, I want to reassure the authors that I do believe a study like this is warranted and I commend the authors for the amount of time and effort that compiling these data must have taken. I also think there is an interesting signal in these data, but the authors need to do a bit more to make their argument more convincing. Maybe my point about the spatial variability is not valid and the sampling distribution is sufficient, but the authors need to demonstrate that convincingly so that their hard work will be rewarded by the community. Thank you for your constructive input to the manuscript. We hope the revisions have addressed your concerns and questions.

REVIEWERS' COMMENTS

Reviewer #1 (Remarks to the Author):

Review of the revisions to "Warm meets fresh: Atlantification of Northeast Greenland Shelf and Fjords"

I thank the authors for thoroughly addressing my comments and the comments from the other reviewer. I also apologize for taking too long to review the revisions, but field work kept me occupied.

In general I think the authors have addressed my concerns. It took me also a while to gauge some of the responses, but I think they are reasonable (with the exception of the title change, see below) even if I might have suggested some aspects differently. A few minor points that could be addressed are listed below, but they are not crucial. I would thus recommend publication of the manuscript using the original title. Given that the paper documents major changes in an important area of ocean-sea ice-glacier ice interaction and that it discusses them well (including hypotheses on the ecosystem ramifications), I am sure that the paper will make an important addition to the literature, i.e. it is appropriate for NatComms.

The original title was "Warm meets fresh: Vertical redistribution of principle water masses on the Northeast Greenland Shelf". Now you have changed the title to "Warm meets fresh: Atlantification of Northeast Greenland Shelf and Fjords" without mentioning this in your rebuttal or the track changed version. Given my very first comment in the original review "The interpretation, however, at times goes way too far beyond what is shown in the manuscript and hence remains unconvincingly speculative.", I have to say that I dislike the change in the title. The change goes in the opposite direction of what I recommended. Rather than reining in the speculations, it doubles down on the speculation and in fact makes it an even more central message that will probably, inappropriately, stick as a result of this manuscript. Hence, I would suggest to stay with the original title, which well represents the scope of the results of the manuscript.

original comment: I248 "more summer sea ice melt": How big is the winter ice volume?

reply: We did not calculate this, since we want to identify the changes in summer only as the hydrography is restricted to summer.

follow up comment: Yes, but the explanation for what you see/describe may also lie in winter! The residence time on the shelf can be long enough. Maybe at least mention that.

I357 of track changed version: "nitracline" not "nitricline"

My comment on original I397 regarding the quality of GEBCO2020. It might be more relevant to look into the Source Identifier Grid (second to last link on https://www.gebco.net/data_and_products/gridded_bathymetry_data/arctic_ocean/) to see whether there are soundings in the fjords and in particular in locations of possible sills/at the fjord entrances. I don't think it's necessary to do this here for this manuscript though it could be mentioned as a way to settle the matter regarding flow restrictions more accurately.

I471 of original manuscript (units of the available potential energy): In the way you present it in the revised version, your APE has units of $J \cdot m^{-4}$, not $J \cdot m^{-3}$ as you state in the text. Not sure why you would choose this definition, but at least keep the units consistent with your definition (or vice versa).

original comment on I482 "Jun/Jul/Aug" Why not "Jul/Aug/Sep", compare I388?

reply: We use JJA rather than JAS to derived sea ice volume transport estimates through Fram Strait as we assume a one-month transit time from Fram Strait into the NEGS.

follow up comment: And where in the text do you state that assumption?

Fig6 caption: "fjords (left) and eastward across the shelf (right of the sill)"

Reviewer #2 (Remarks to the Author):

Review of "Warm meets fresh: Vertical redistribution of principle water masses on the Northeast Greenland Shelf" by Gjelstrup et al.

I found the revised manuscript to be much improved, and I recommend it be accepted pending (very) minor revisions listed below. The authors did a wonderful job responding to the two reviewers' comments, and the manuscript has clearly benefited from these edits. The rewritten introduction allows the reader to more easily understand the motivation for the study, and the discussion about accounting for inhomogeneous sampling is well-reasoned. I also appreciated the APE citations for my own edification.

Minor edits:

Line 45 – insert a comma after "NEGS"

Line 120 – replace "accounted for" with "addressed"

Line 321 – there appears to be a period before "During"

Line 392 – there appears to be a period after "per". I recommend rewriting to salinity m^{-1} to be consistent with the rest of the text.

Reply to review

Warm meets fresh: vertical redistribution of principle water masses on the Northeast Greenland Shelf

We once again thank both the reviewers for their comments. Please find a point-by-point response to all points raised below. Reviewer comments are highlighted in grey and our response is below.

Reviewer #1 (Remarks to the Author):

The original title was “Warm meets fresh: Vertical redistribution of principle water masses on the Northeast Greenland Shelf”. Now you have changed the title to “Warm meets fresh: Atlantification of Northeast Greenland Shelf and Fjords” without mentioning this in your rebuttal or the track changed version. Given my very first comment in the original review “The interpretation, however, at times goes way too far beyond what is shown in the manuscript and hence remains unconvincingly speculative.”, I have to say that I dislike the change in the title. The change goes in the opposite direction of what I recommended. Rather than reining in the speculations, it doubles down on the speculation and in fact makes it an even more central message that will probably, inappropriately, stick as a result of this manuscript. Hence, I would suggest to stay with the original title, which well represents the scope of the results of the manuscript.

Agreed. We have returned to the original title for the final version of the manuscript.

original comment: l248 “more summer sea ice melt”: How big is the winter ice volume?

reply: We did not calculate this, since we want to identify the changes in summer only as the hydrography is restricted to summer.

follow up comment: Yes, but the explanation for what you see/describe may also lie in winter! The residence time on the shelf can be long enough. Maybe at least mention that.

A possible contribution from winter sea ice is now explicitly acknowledged in the text (line 271 tracked changes version).

l357 of track changed version: “nitracline” not “nitricline”

Corrected.

My comment on original l397 regarding the quality of GEBCO2020. It might be more relevant to look into the Source Identifier Grid (second to last link on https://www.gebco.net/data_and_products/gridded_bathymetry_data/arctic_ocean/) to see whether there are soundings in the fjords and in particular in locations of possible sills/at the fjord entrances. I don't think it's necessary to do this here for this manuscript though it could be mentioned as a way to settle the matter regarding flow restrictions more accurately.

Following the reviewers recommendation, fjord soundings were sought after but to no avail. Therefore, no modifications have been made.

l471 of original manuscript (units of the available potential energy): In the way you present it in the revised version, your APE has units of $J \cdot m^{-4}$, not $J \cdot m^{-3}$ as you state in the text. Not sure why you would choose this definition, but at least keep the units consistent with your definition (or vice versa).

Neither the definition nor units of available potential energy have changed during revision of manuscript. Please see <https://link.springer.com/article/10.1007/s00382-019-04816-y>, which uses a similar definition to the one used in the current manuscript. To avoid any further confusion we also derive the units below.

$$APE = \frac{1}{h} \cdot \int_{-z_2}^{-z_1} g \cdot \hat{\rho} \cdot z \, dz$$

Units:

$$\frac{1}{m} \cdot \frac{m}{s^2} \cdot \frac{kg}{m^3} \cdot \frac{m}{1} \cdot \frac{m}{1} = \frac{kg \cdot m^3}{s^2 \cdot m^4} = \frac{J}{m^3}$$

original comment on l482 “Jun/Jul/Aug” Why not “Jul/Aug/Sep”, compare l388?

reply: We use JJA rather than JAS to derived sea ice volume transport estimates through Fram Strait as we

assume a one-month transit time from Fram Strait into the NEGS.
follow up comment: And where in the text do you state that assumption?
This assumption is stated explicitly in the main text on lines 263-264 (tracked changes version).

Fig6 caption: "fjords (left) and eastward across the shelf (right of the sill)"
Corrected.

Reviewer #2 (Remarks to the Author):

Line 45 – insert a comma after "NEGS"
Done.

Line 120 – replace "accounted for" with "addressed".
Corrected.

Line 321 – there appears to be a period before "During"
Corrected.

Line 392 – there appears to be a period after "per". I recommend rewriting to salinity m-1 to be consistent with the rest of the text.
The recommendation has been implemented.